# FreeDyG: Frequency Enhanced Continuous-Time Dynamic Graph Model for Link Prediction

**Yuxing Tian**[1][*][†] **Yiyan Qi**[1][*][‡] **FanGuo**[2]
[1]IDEA Research, International Digital Economy Academy
[2]Jiangxi Normal University

## Abstract

Link prediction is a crucial task in dynamic graph learning. Recent advancements in continuous-time dynamic graph models, primarily by leveraging richer temporal details, have significantly improved link prediction performance. However, due to their complex modules, they still face several challenges, such as overfitting and optimization difficulties. More importantly, it is challenging for these methods to capture the 'shift' phenomenon, where node interaction patterns change over time. To address these issues, we propose a simple yet novel method called **Fr**equency **E**nhanced Continuous-Time **Dy**namic **G**raph (**FreeDyG**) model for link prediction. Specifically, we propose a node interaction frequency encoding module that both explicitly captures the proportion of common neighbors and the frequency of the interaction of the node pair. Unlike previous works that primarily focus on the time domain, we delve into the frequency domain, allowing a deeper and more nuanced extraction of interaction patterns, revealing periodic and "shift" behaviors. Extensive experiments conducted on seven real-world continuous-time dynamic graph datasets validate the effectiveness of FreeDyG. The results consistently demonstrate that FreeDyG outperforms existing methods in both transductive and inductive settings. Our code is available at this repository: https://github.com/Tianxzzz/FreeDyG

## 1 Introduction

Link prediction on dynamic graphs is a fundamental problem in various real-world applications, such as social media analysis Huo et al. (2018); Alvarez-Rodriguez et al. (2021), recommendation systems Song et al. (2019); Yuxiao et al. (2012); Wang et al. (2021b), and drug discovery Abbas et al. (2021). Recently, dynamic graph representation learning methods have become widespread in both industry and academia due to their remarkable performance in solving this problem.

Existing dynamic graph learning methods can be divided into two main categories: discrete-time dynamic graph (DTDG) models Pareja et al. (2020); Zhao et al. (2019); Yang et al. and continuous-time dynamic graph (CTDG) models Kumar et al. (2019); Wang et al. (2021c); Xu et al. (2020); Trivedi et al. (2019); Yu et al. (2023); Wang et al. (2021a;d); Jin et al. (2022a); Luo et al. (2023); Chang et al. (2020); Huang et al. (2020). Notably, there has been a growing interest in CTDG algorithms, primarily because of their ability to preserve time information effectively.

Although the above CTDG methods have achieved impressive results, they still have some limitations. Firstly, some methods rely on random walks (RW) Wang et al. (2021d); Jin et al. (2022b), temporal point processes (TPP) Chang et al. (2020); Huang et al. (2020), or ordinary differential equations (ODE) Luo et al. (2023); Liang et al. (2022) to enhance their learning ability. However, these methods are computationally expensive and may lead to overfitting of the historical interactions. Secondly, most existing methods encode interaction sequences independently, disregarding potential relationships between them. Although Yu et al. (2023) proposes the neighbor

---

[*]Equal contribution.
[†]Work done during internship at IDEA Research.
[‡]Corresponding author.

co-occurrence mechanism, it relies solely on the number of historical common neighbors between node pairs to predict the likelihood of node pairs interacting in the future. As a result, it is difficult for them to capture the "shift" phenomenon which is commonly hidden behind time series data. For instance, in a co-author network, researchers who are acquainted with each other tend to collaborate during the same period each year, as conferences are typically held around the same time annually. However, when a new and popular research topic emerges suddenly, such as large language models (LLMs), there is a notable shift in these collaboration patterns. Researchers then show a preference for establishing collaborations with unfamiliar researchers who specialize in that area.

To address the above challenges, we delve into the frequency domain and propose a simple yet novel method called **Fre**quency **E**nhanced Continuous-Time **Dy**namic **G**raph (**FreeDyG**) model for link prediction. FreeDyG comprises two essential components: the encoding layer and the frequency-enhanced MLP-Mixer layer. The encoding layer is designed to transform each interaction $(u, v, t)$ into an embedding sequence, considering the historical neighbors of both node $u$ and $v$. In contrast to existing methods, we also introduce a new node interaction frequency encoding approach to explicitly capture the interaction frequency between node pairs. Then, we propose a novel frequency-enhanced MLP-Mixer layer to efficiently capture the periodic temporal patterns and the "shift" phenomenon hidden in the frequency domain. Specifically, we apply the Fourier Transform to the encoded embedding sequence, followed by multiplication with a learnable complex number tensor. This can adaptively enhance desired frequencies, thereby improving the model's ability to capture relevant information.

Our contributions can be summarized as follows: Firstly, we introduce a novel frequency-enhanced dynamic graph model called FreeDyG for the task of link prediction. Instead of the temporal domain, FreeDyG tries to address this problem by delving into the frequency domain. Secondly, in addition to the traditional encoding of node/link features and time information, we propose a node interaction frequency encoding approach that explicitly captures the frequency of neighbor nodes from both the source and target nodes. Furthermore, we design a novel frequency-enhanced MLP-Mixer layer to further capture periodic temporal patterns and the "shift" phenomenon present in the frequency domain. Finally, we conduct extensive experiments on seven widely used real-world continuous-time dynamic graph datasets, evaluating both transductive and inductive settings. The experimental results demonstrate the superior performance of FreeDyG compared to state-of-the-art methods.

## 2 PRELIMINARIES

**Task Definition.** Following prevailing methods Poursafaei et al. (2022); Yu et al. (2023); Rossi et al. (2020), our study primarily focuses on CTDGs with edge addition events. A CTDG $G$ can be represented as a chronological sequence of interactions between specific node pairs: $G = \{(u_0, v_0, t_0), (u_1, v_1, t_1), \ldots, (u_n, v_n, t_n)\}$, where $t_i$ denotes the timestamp and the timestamps are ordered as $(0 \leq t_0 \leq t_1 \leq \ldots \leq t_n)$. $u_i, v_i \in V$ denote the node IDs of the $i - th$ interaction at timestamp $t_i$, $V$ is the entire node set. Each node $u \in V$ is associated with node feature $x_u \in R^{d_N}$, and each interaction $(u, v, t)$ has edge feature $e_{u,v}^t \in R^{d_E}$, where $d_N$ and $d_E$ denote the dimensions of the node and link feature respectively. Based on the above definitions, link prediction in CTDGs can be formulated as: given a pair of nodes with a specific timestamp $t$, we aim to predict whether the two nodes are connected at $t$ based on all the available historical data.

**Discrete Fourier Transform (DFT).** Given a sequence of data $\{x_n\}_{n=1}^N$, the Discrete Fourier Transform (DFT) is utilized to convert the sequence into the frequency domain. The DFT operation can be defined as:

$$X_k = \sum_{k=1}^N x_n e^{-\frac{2\pi i}{N} nk}, \quad 1 \leq k \leq N \tag{1}$$

where $i$ denotes the imaginary unit and $\{X_k\}_{k=1}^N$ is a sequence of complex numbers in the frequency domain. Thus, $X_k$ captures the spectrum of the sequence $\{x_n\}_{n=1}^N$ at the frequency $\omega_k = 2\pi k/M$. We note that the DFT is a one-to-one transformation, enabling the recovery of the original sequence through the inverse DFT (IDFT):

$$x_n = \frac{1}{N} \sum_{k=1}^N X_k e^{\frac{2\pi i}{N} nk} \tag{2}$$

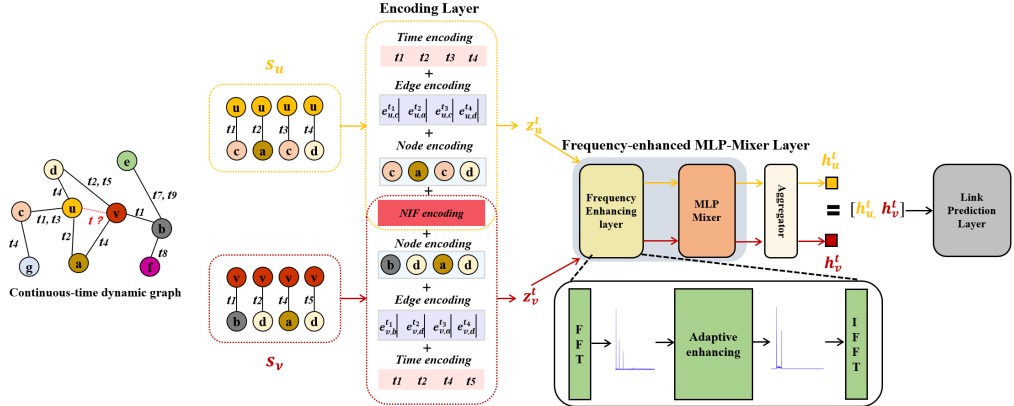

Figure 1: The overview of FreeDyG.

The computation complexity of DFT is $O(N^2)$. In practice, the Fast Fourier Transform (FFT) is commonly used to compute the DFT efficiently, which could recursively express the DFT of a sequence of length $N$ and reduce the computation complexity from $O(N^2)$ to $O(N \log N)$. Similarly, the IDFT in Equation 2 can be efficiently computed using the inverse Fast Fourier Transform (IFFT). Due to the page limit, we omit the details of DFT and FFT, which can be found in Bruun (1978).

## 3  METHOLODGY

In this section, we provide a detailed description of our FreeDyG. As shown in Figure 1, the model operates on a node pair $(u, v)$ and a specific timestamp $(t)$. Initially, we sample $L$ first-hop historical neighbors for both nodes, ordered based on temporal proximity, to construct two interaction sequences, $S_u^t$ and $S_v^t$. If a node has fewer than $L$ historical neighbors, zero-padding is used to fill the gap. Subsequently, we employ an encoding layer to encode the features of each node, link, and timestamp within the sequences. Additionally, we incorporate the frequencies of neighbor appearances in both $S_u^t$ and $S_v^t$ to exploit correlations between nodes. The encoded representations are then aligned and fed into a frequency-enhanced MLP-Mixer layer, enabling the capture of information at different frequencies. The outputs are aggregated to generate time-aware representations of $u$ and $v$ at timestamp $t$ (i.e., $h_u^t$ and $h_v^t$). The final prediction is generated by the link prediction layer.

### 3.1  ENCODING LAYER

**Node/Edge Encoding.** In dynamic graphs, both nodes and edges (links) frequently possess associated features. To derive embeddings for interactions, it is sufficient to extract the intrinsic features of the neighboring nodes and edges based on the sequence $S_*^t$, where $*$ can be either $u$ or $v$. Similar to existing works, we encode the nodes and links as $Z_{*,N}^t \in \mathbb{R}^{L \times d_N}$ and $Z_{*,E}^t \in \mathbb{R}^{L \times d_E}$ respectively, where $d_N, d_E$ are the dimensions of node and edge embeddings respectively.

**Time Encoding.** Time encoding is employed to map constant timestamps to vectors $Z_{*,T}^t \in \mathbb{R}^{L \times d_T}$, where $d_T$ is the dimension of time embeddings. Specifically, we utilize the widely adopted time encoding function $cos(t_n\omega)$, where $\boldsymbol{\omega} = \left\{\alpha^{-(i-1)/\beta}\right\}_{i=1}^{d_T}$ is employed to encode timestamps. $\alpha$ and $\beta$ are hyperparameters to make $t_{max} \times \alpha^{-(i-1)/\beta}$ close to 0 when i close to $d_T$. A cosine function is then applied to project these values into the range of $[-1, +1]$. Notably, we use relative timestamps instead of absolute timestamps for encoding. In other words, if the timestamp of the sampled interaction is $t_0$, and the specific timestamp for link prediction is $t$, we utilize $cos((t_n-t)\omega)$ as the effective relative time encoding function. It is worth mentioning that $\omega$ remains constant and is not updated during the training phase. This technique makes the model easy to optimize and leads to performance improvement, as demonstrated in Cong et al. (2023).

**Node Interaction Frequency (NIF) Encoding.** Most existing methods encode interaction sequences, i.e., $S_u^t$ and $S_v^t$, independently, disregarding potential relationships between them. Yu et al. (2023) introduces a neighbor co-occurrence scheme, suggesting that nodes with a higher number of shared historical neighbors are more likely to interact in the future. However, while a significant portion of edges in networks exhibit recurring patterns over time, these recurrence patterns vary considerably across different networks and domains Poursafaei et al. (2022).

To address this issue, we introduce a node interaction frequency encoding approach, which takes into account not only the appearance frequency of common neighbors but also the interaction frequency between the node pairs. It recognizes that the potential for future interactions between two nodes is influenced by both their common neighbors and their past interactions. Specifically, given the interaction sequences $S_u^t$ and $S_v^t$, we count the frequency of each neighbor in both $S_u^t$ and $S_v^t$. In addition, We specifically encode for the frequency of the interaction of the node pair and get the node interaction frequency features for $u$ and $v$, which are represented by $Z_{*,C}^t$. For example, suppose the historical interacted nodes of $u$ and $v$ are $\{c, v, c, d, v\}$ and $\{d, u, d, c, u\}$. The appearing frequencies of $c$, $d$ in $u/v$'s historical interactions are $2/1$, $1/2$ respectively. And the node pair interaction frequency is $[2, 2]$. Therefore, the node interaction frequency features of $u$ and $v$ can be denoted by $F_u^t = [[2, 1], [2, 2], [2, 1], [1, 2], [2, 2]]^T$ and $F_v^t = [[1, 2], [2, 2], [1, 2], [2, 1], [2, 2]]^T$. Then, we encode the node interaction frequency features by:

$$Z_{*,F}^t = f\left(F_*^t[:, 0]\right) + f\left(F_*^t[:, 1]\right) \in \mathbb{R}^{L \times d_F}, \tag{3}$$

where $*$ could be $u$ or $v$. $f()$ represents a two-layer perceptron with $ReLU$ activation. And the dimension of input and output of $f()$ are 1 and $d_F$.

Finally, we concatenate all encodings mentioned above to an embedding of dimension $d$ with trainable weight $W_* \in R^{d_* \times d}$ and $b_* \in R^d$, resulting in $Z_{u,*}^t \in R^{l_u^t \times d}$ and $Z_{v,*}^t \in R^{l_v^t \times d}$. Here, $*$ can be $N$, $E$, $T$ or $F$. Subsequently, the concatenated encodings for nodes are summed as $Z_u^t \in \mathbb{R}^{L \times d} = Z_{u,N}^t + Z_{u,E}^t + Z_{u,T}^t + Z_{u,F}^t$ and $Z_v^t \in \mathbb{R}^{L \times d} = Z_{v,N}^t + Z_{v,E}^t + Z_{v,T}^t + Z_{v,F}^t$, respectively.

## 3.2 FREQUENCY-ENHANCED MLP-MIXER LAYER

We derive the historical interaction embedding sequences for the node pair by utilizing the encoding layer. Since each interaction embedding is arranged chronologically, it can be viewed as discrete time series data. Consequently, we naturally leverage the Fourier transform to decompose of time series data into their constituent frequencies, which can effectively capture the interaction patterns of nodes across various frequencies. To this end, we introduce the frequency-enhanced MLP-Mixer layer, which contains two sublayers: the frequency-enhancing layer and the MLP-Mixer layer.

**Frequency Enhancing (FE) layer.** Given the input $Z_*^l$ of the $l$-th layer and the first layer input is $Z_*^t$. For simplicity, $Z_*^l$ is short for $Z_*^{l,t}$. To better identify important frequencies within the historical interaction sequence data, we perform the Fast Fourier Transform (FFT) along the first dimension of $Z_*^l \in \mathbb{R}^{L \times d}$ to convert it into the frequency domain:

$$\mathcal{Z}_*^l = \mathcal{F}(Z_*^l), \tag{4}$$

where $\mathcal{F}$ denotes the 1D FFT. $\mathcal{Z}_*^l \in \mathbb{C}^{\{\frac{L}{2}+1\} \times d}$ denotes the frequency components of $Z_*^l$ and $\mathbb{C}$ denotes the complex number domain. Then we can adaptively enhance the frequency components by multiplying it with a learnable complex number tensor $\mathcal{W} \in \mathbb{C}^{\{\frac{L}{2}+1\} \times d}$.

$$\widehat{\mathcal{Z}}_*^l = \mathcal{W} \cdot \mathcal{Z}_*^l \tag{5}$$

where $\cdot$ denotes the element-wise multiplication, $\widehat{\mathcal{Z}}_*^l \in \mathbb{C}^{\{\frac{L}{2}+1\} \times d}$ represents the enhanced frequency components. Finally, we transform $\widehat{\mathcal{Z}}_*^l$ back to the time domain.

$$\widetilde{Z}_*^l \leftarrow \mathcal{F}^{-1}\left(\widehat{\mathcal{Z}}_*^l\right). \tag{6}$$

where $\mathcal{F}^{-1}()$ denotes the inverse 1D FFT, which converts the complex number tensor into a real number tensor. Then we use the residual connection and dropout layer as:

$$Z_*^l = Z_*^l + \text{Dropout}\left(\widetilde{Z}_*^l\right) \tag{7}$$

**MLP-Mixer layer.** After that, we employ MLP-Mixer Tolstikhin et al. (2021) to further capture the non-linearity characteristics, which contains two types of layers: token mixing MLP layer and channel mixing MLP layer. The computation is listed as:

$$\begin{aligned} Z_{*,\text{token}}^l &= Z_*^l + \mathbf{W}_{\text{token}}^2 \, \sigma(\mathbf{W}_{\text{token}}^1 \, \text{LayerNorm}(Z_*^l)), \\ Z_{*,\text{channel}}^l &= Z_{*,\text{token}}^l + \sigma(\text{LayerNorm}(Z_{*,\text{token}}^l)\mathbf{W}_{\text{channel}}^1)\mathbf{W}_{\text{channel}}^2 \end{aligned} \tag{8}$$

where $\sigma$ is an element-wise nonlinearity, $\mathbf{W}^*_{\text{token}} \in \mathbb{R}^{d \times d}$ and $\mathbf{W}^*_{\text{channel}} \in \mathbb{R}^{d \times d}$ are trainable parameters in the token-mixing and channel-mixing MLPs, respectively. The same channel-mixing MLP layer (token-mixing MLP layer) is applied to every row (column) of input.

**Theorem 1.** *Given an input $Z \in \mathbb{R}^{N \times D}$ and a learnable matrix $\mathcal{W} \in \mathbb{R}^{N \times D}$ and their corresponding frequency domain conversion, the multiplication in the frequency domain is equivalent to the global convolution in the time domain.*

**Proof.** We regard $Z \in \mathbb{R}^{N \times D}$ as $\{z_n \in \mathbb{R}^D\}_{n=1}^N$ and $\mathcal{W} \in \mathbb{R}^{N \times D}$ as $\{w_n \in \mathbb{R}^D\}_{n=1}^N$. Denote the $d$-th dimension features of $Z$ and $W$ as $\{z_n^{(d)}\}_{n=1}^N$ and $\{w_n^{(d)}\}_{n=1}^N$. Then, the global convolution can be defined as:

$$\{z_n^{(d)}\}_{n=1}^N \circledast \{w_n^{(d)}\}_{n=1}^N = \sum_{m=1}^N w_m^{(d)} \cdot z_{(n-m) \bmod N}^{(d)} \tag{9}$$

where $\circledast$ and $\bmod$ denote the convolution and integer modulo operation, respectively. Then, according to Equation 1, the multiplication in the frequency domain can be written as:

$$
\begin{aligned}
\mathcal{DFT}(\{w_n^{(d)}\}_{n=1}^N) \cdot \mathcal{DFT}(\{z_n^{(d)}\}_{n=1}^N) &= w_k^{(d)} \sum_{n=1}^N z_n^{(d)} e^{-\frac{2\pi i}{N} kn} \\
&= w_k^{(d)} \left( \sum_{n=1}^{N-m} z_n^{(d)} e^{-\frac{2\pi i}{N} kn} + \sum_{n=N-m}^N z_n^{(d)} e^{-\frac{2\pi i}{N} kn} \right) \\
&= w_k^{(d)} \left( \sum_{n=m}^N z_{n-m}^{(d)} e^{-\frac{2\pi i}{N} k(n-m)} + \sum_{n=1}^m z_{n-m+N}^{(d)} e^{-\frac{2\pi i}{N} k(n-m)} \right) \\
&= \sum_{m=1}^N w_m^{(d)} e^{-\frac{2\pi i}{N} km} \sum_{n=1}^N z_{(n-m)\%N}^{(d)} e^{-\frac{2\pi i}{N} k(n-m)} \\
&= \mathcal{DFT}(\{z_n^{(d)}\}_{n=1}^N \circledast \{w_n^{(d)}\}_{n=1}^N)
\end{aligned}
\tag{10}
$$

## 3.3 AGGREGATOR

Different from most previous methods that simply aggregate the sequence of each appearing element using average pooling, we get the time-aware representations $h_u^t \in \mathbb{R}^d$ and $h_v^t \in \mathbb{R}^d$ of node $u$ and $v$ at timestamp $t$ by the following weighted aggregation equation:

$$h_*^t = ((W^{agg} \cdot Z_{*,\text{channel}}^l)^T \cdot Z_{*,\text{channel}}^l)^T \tag{11}$$

where $W^{agg} \in \mathbb{R}^{1 \times L}$ is a trainable vector designed to adaptively learn the importance of various interactions.

## 3.4 LINK PREDICTION LAYER

The prediction $\hat{y}$ is computed by adopting a 2-layer multilayer perceptron (MLP) and using softmax to convert it into link probability on the concatenation of the above two nodes' representation:

$$\hat{y} = Softmax(MLP(RELU(MLP(h_u^t || h_v^t)))). \tag{12}$$

## 3.5 LOSS FUNCTION

For link prediction loss, we adopt binary cross-entropy loss function, which is defined as:

$$\mathcal{L}_p = -\frac{1}{K} \sum_{i=1}^S (y_i * log\hat{y_i} + (1 - y_i) * log(1 - \hat{y_i})), \tag{13}$$

where $K$ is the number of positive/negative samples, $y_i$ represents the ground-truth label of $i$-th sample and the $\hat{y_i}$ represents the prediction value.

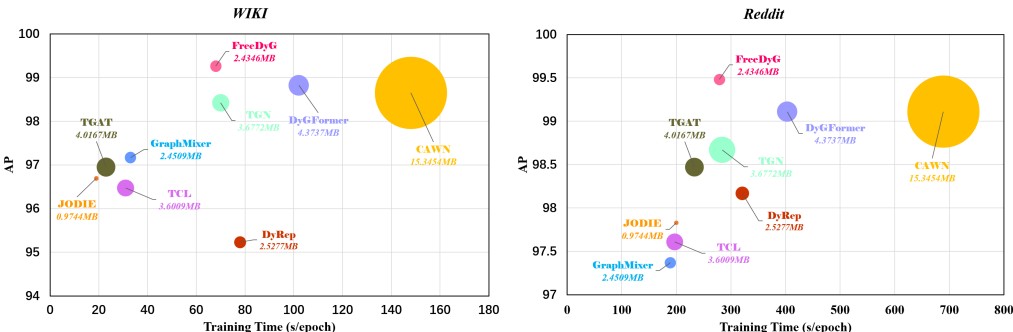

Figure 2: Comparison of model performance, parameter size and training time per epoch on WIKI and Reddit.

## 4 EXPERIMENTS

### 4.1 DATASETS

We utilize seven publicly available real-world datasets: Wiki, REDDIT, MOOC, LastFM, Enron, Social Evo, and UCI, in our study. Appendix A provides a detailed description and the statistics of the datasets are shown in Table 4. The sparsity of the graphs is quantified using the density score, calculated as $\frac{2|E|}{|V|(|V|-1)}$, where $|E|$ and $|V|$ represent the number of links and nodes in the training set, respectively. To facilitate training, validation, and testing, we split these datasets into three chronological segments with ratios of 70%-15%-15%.

### 4.2 BASELINES

To evaluate the performance of our method, we conduct experiments comparing it with previous state-of-the-arts, including JODIE (Kumar et al., 2019), DyRep (Trivedi et al., 2019), TGAT (Xu et al., 2020), TGN (Rossi et al., 2020), CAWN (Wang et al., 2021d), EdgeBank (Poursafaei et al., 2022), TCL (Wang et al., 2021a), GraphMixer (Cong et al., 2023), DyGFormer (Yu et al., 2023). Detailed descriptions of these methods can be found in Appendix B.

### 4.3 EVALUATION METRICS AND EXPERIMENTAL SETTINGS

For evaluating our method, we employ Average Precision (AP) and Area Under the Receiver Operating Characteristic Curve (AUC-ROC) as the evaluation metrics. The link prediction task encompasses two settings: 1) **transductive setting**, which focuses on predicting future links between nodes observed during training, and 2) **inductive setting**, which involves predicting future links between unseen nodes. To ensure comprehensive comparisons, we also evaluate each method with three negative sampling strategies as Poursafaei et al. (2022), i.e., random (rnd), historical (hist), and inductive (ind) negative sampling, which the latter two are more challenging. The evaluation details of all three strategies can be found in Appendix C.

All models are trained for a maximum of 200 epochs using the early stopping strategy with patience of 20. The model that achieves the highest performance on the validation set is selected for testing. For all models, we employ the Adam optimizer and set the learning rate and batch size to 0.0001 and 200, respectively. The hyperparameter configurations of the baselines align with those specified in their respective papers. For our FreeDyG, we set the $d_T$ to 100, and both $\alpha$ and $\beta$ to 10. The number of frequency-enhanced MLP-Mixer layers are 2. We conduct ten runs of each method with different seeds and report the average performance to eliminate deviations. All experiments are performed on an NVIDIA A100-SXM4 40GB GPU.

### 4.4 COMPARISON WITH SOTA

In this section, we compare our FreeDyG with the previous SOTA in both transductive and inductive settings. Table 10 and Table 2 show the AP of all datasets in these two settings respectively. To provide a more comprehensive study of our FreeDyG, we show results among all three negative sampling strategies. Due to the limitation of space, we put similar results on AUC-ROC in Table 5 and Table 6 in Appendix. We note that EdgeBank Poursafaei et al. (2022) is only designed for the

| NSS | Datasets | JODIE | DyRep | TGAT | TGN | CAWN | EdgeBank | TCL | GraphMixer | DyGFormer | FreeDyG |
|---|---|---|---|---|---|---|---|---|---|---|---|
| rnd | Wiki | 96.69 ± 0.25 | 95.23 ± 0.50 | 96.95 ± 0.17 | 98.42 ± 0.05 | 98.65 ± 0.04 | 90.37 ± 0.00 | 96.47 ± 0.16 | 97.17 ± 0.05 | 98.82 ± 0.02 | **99.26 ± 0.01** |
| | Reddit | 97.83±0.21 | 98.17±0.02 | 98.47±0.03 | 98.67±0.04 | 99.11±0.01 | 94.86±0.00 | 97.61±0.03 | 97.37±0.01 | 99.11±0.02 | **99.48±0.01** |
| | MOOC | 77.20±1.32 | 79.97±0.82 | 85.44±0.76 | 89.43±2.95 | 78.66±0.31 | 57.97±0.00 | 81.12±0.43 | 82.73±0.16 | 87.23±0.45 | **89.61±0.19** |
| | LastFM | 68.54±2.95 | 70.79±1.87 | 73.76±0.45 | 78.69±2.71 | 86.58±0.10 | 79.29±0.00 | 65.64±2.52 | 75.64±0.23 | 92.07±0.28 | **92.15±0.16** |
| | Enron | 79.10±0.85 | 82.02±3.07 | 72.58±0.79 | 85.33±1.05 | 89.56±0.09 | 83.53±0.00 | 79.70±0.71 | 81.08±0.73 | 92.47±0.12 | **92.51±0.05** |
| | Social Evo. | 88.12±0.74 | 88.87±0.30 | 93.16±0.17 | 93.57±0.17 | 84.96±0.09 | 74.95±0.00 | 93.13±0.16 | 93.37±0.07 | 94.73±0.01 | **94.91±0.01** |
| | UCI | 87.65±1.85 | 70.24±0.32 | 79.55±0.83 | 90.69±0.45 | 94.35±0.11 | 76.20±0.00 | 88.12±2.73 | 93.50±0.49 | 95.76±0.15 | **96.28±0.11** |
| **Avg.Rank** | | 7.14 | 6.08 | 5.85 | 3.31 | 2.94 | 8.79 | 4.56 | 5.23 | 2.14 | **1** |
| hist | Wiki | 81.19±0.48 | 78.32±0.71 | 87.01±0.19 | 86.96±0.36 | 72.38±1.85 | 73.35 ± 0.00 | 88.75±0.27 | 90.87±0.08 | 82.23 ± 2.54 | **91.59±0.57** |
| | Reddit | 80.03 ± 0.36 | 79.83 ± 0.31 | 79.55 ± 0.20 | 81.75 ± 0.36 | 80.82 ± 0.45 | 73.59 ± 0.00 | 77.14 ± 0.16 | 78.44 ± 0.18 | 81.02 ± 0.59 | **85.67 ± 1.01** |
| | MOOC | 78.94 ± 1.25 | 75.60 ± 1.12 | 82.19 ± 0.62 | **87.06 ± 1.93** | 74.05 ± 0.95 | 60.71 ± 0.00 | 77.06 ± 0.41 | 77.77 ± 0.92 | 85.85 ± 0.66 | 86.71 ± 0.81 |
| | LastFM | 74.35 ± 3.81 | 74.92 ± 2.46 | 71.59 ± 0.24 | 76.87 ± 4.64 | 69.86 ± 0.43 | 73.03 ± 0.00 | 59.30 ± 2.31 | 72.47 ± 0.49 | **81.57 ± 0.48** | 79.71 ± 0.51 |
| | Enron | 69.85 ± 2.70 | 71.19 ± 2.76 | 64.07 ± 1.05 | 73.91 ± 1.76 | 64.73 ± 0.36 | 76.53 ± 0.00 | 70.66 ± 0.39 | 77.98 ± 0.92 | 75.63 ± 0.73 | **78.87 ± 0.82** |
| | Social Evo. | 87.44 ± 6.78 | 93.29 ± 0.43 | 95.01 ± 0.44 | 94.45 ± 0.56 | 85.53 ± 0.38 | 80.57 ± 0.00 | 94.74 ± 0.31 | 94.93 ± 0.31 | 97.38 ± 0.14 | **97.79 ± 0.23** |
| | UCI | 75.24 ± 5.80 | 55.10 ± 3.14 | 68.27 ± 1.37 | 80.43 ± 2.12 | 65.30 ± 0.43 | 65.50 ± 0.00 | 80.25 ± 2.74 | 84.11 ± 1.35 | 82.17 ± 0.82 | **86.10 ± 1.19** |
| **Avg.Rank** | | 5.46 | 5.08 | 5.08 | 3.85 | 7.54 | 5.92 | 5.46 | 4.00 | 2.85 | **1.28** |
| ind | Wiki | 75.65 ± 0.79 | 70.21 ± 1.58 | 87.00 ± 0.16 | 85.62 ± 0.44 | 74.06 ± 2.62 | 80.63 ± 0.00 | 86.76 ± 0.72 | 88.59 ± 0.17 | 78.29 ± 5.38 | **90.05 ± 0.79** |
| | Reddit | 86.98 ± 0.16 | 86.30 ± 0.26 | 89.59 ± 0.24 | 88.10 ± 0.24 | **91.67 ± 0.24** | 85.48 ± 0.00 | 87.45 ± 0.29 | 85.26 ± 0.11 | 91.11 ± 0.40 | 90.74 ± 0.17 |
| | MOOC | 65.23 ± 2.19 | 61.66 ± 0.95 | 75.95 ± 0.64 | 77.50 ± 2.91 | 73.51 ± 0.94 | 49.43 ± 0.00 | 74.65 ± 0.54 | 74.27 ± 0.92 | 81.24 ± 0.69 | **83.01 ± 0.87** |
| | LastFM | 62.67 ± 4.49 | 64.41 ± 2.70 | 71.13 ± 0.17 | 65.95 ± 5.98 | 67.48 ± 0.77 | **75.49 ± 0.00** | 58.21 ± 0.89 | 68.12 ± 0.33 | 73.97 ± 0.50 | 72.19 ± 0.24 |
| | Enron | 68.96 ± 0.98 | 67.79 ± 1.53 | 63.94 ± 1.36 | 70.89 ± 2.72 | 75.15 ± 0.58 | 73.89 ± 0.00 | 71.29 ± 0.32 | 75.01 ± 0.79 | 77.41 ± 0.89 | **77.81 ± 0.65** |
| | Social Evo. | 89.82 ± 4.11 | 93.28 ± 0.48 | 94.84 ± 0.44 | 95.13 ± 0.56 | 88.32 ± 0.27 | 83.69 ± 0.00 | 94.90 ± 0.36 | 94.72 ± 0.33 | **97.68 ± 0.10** | 97.57 ± 0.15 |
| | UCI | 65.99 ± 1.40 | 54.79 ± 1.76 | 68.67 ± 0.84 | 70.94 ± 0.71 | 64.61 ± 0.48 | 57.43 ± 0.00 | 76.01 ± 1.11 | 80.10 ± 0.51 | 72.25 ± 1.71 | **82.35 ± 0.73** |
| **Avg.Rank** | | 6.62 | 6.38 | 4.15 | 4.38 | 5.46 | 5.62 | 4.69 | 4.46 | 2.43 | **1.71** |

Table 1: AP for transductive link prediction with three different negative sampling strategies.

| NSS | Datasets | JODIE | DyRep | TGAT | TGN | CAWN | TCL | GraphMixer | DyGFormer | FreeDyG |
|---|---|---|---|---|---|---|---|---|---|---|
| rnd | Wikipedia | 94.82 ± 0.20 | 92.43 ± 0.37 | 96.22 ± 0.07 | 97.83 ± 0.04 | 98.24 ± 0.03 | 96.22 ± 0.17 | 96.65 ± 0.02 | 98.59 ± 0.03 | **98.97±0.01** |
| | Reddit | 96.50 ± 0.13 | 96.09 ± 0.11 | 97.09 ± 0.04 | 97.50 ± 0.07 | 98.62 ± 0.01 | 94.09 ± 0.07 | 95.26 ± 0.02 | 98.84 ± 0.02 | **98.91 ± 0.01** |
| | MOOC | 79.63 ± 1.92 | 81.07 ± 0.44 | 85.50 ± 0.19 | **89.04 ± 1.17** | 81.42 ± 0.24 | 80.60 ± 0.22 | 81.41 ± 0.21 | 86.96 ± 0.43 | 87.75± 0.62 |
| | LastFM | 81.61 ± 3.82 | 83.02 ± 1.48 | 78.63 ± 0.31 | 81.45 ± 4.29 | 89.42 ± 0.07 | 73.53 ± 1.66 | 82.11 ± 0.42 | 94.23 ± 0.09 | **94.89 ± 0.01** |
| | Enron | 80.72 ± 1.39 | 74.55 ± 3.95 | 67.05 ± 1.51 | 77.94 ± 1.02 | 86.35 ± 0.51 | 76.14 ± 0.79 | 75.88 ± 0.48 | **89.76 ± 0.34** | 89.69 ± 0.17 |
| | Social Evo. | 91.96 ± 0.48 | 90.04 ± 0.47 | 91.41 ± 0.16 | 90.77 ± 0.86 | 79.94 ± 0.18 | 91.55 ± 0.09 | 91.86 ± 0.06 | 93.14 ± 0.04 | **94.76 ± 0.05** |
| | UCI | 79.86 ± 1.48 | 57.48 ± 1.87 | 79.54 ± 0.48 | 88.12 ± 2.05 | 92.73 ± 0.06 | 87.36 ± 2.03 | 91.19 ± 0.42 | 94.54 ± 0.12 | **94.85 ± 0.10** |
| **Avg.Rank** | | 6.14 | 6.14 | 5.45 | 3.71 | 3.14 | 4.56 | 4.45 | 1.71 | **1.14** |
| hist | Wikipedia | 68.69 ± 0.39 | 62.18 ± 1.27 | 84.17 ± 0.22 | 81.76 ± 0.32 | 67.27 ± 1.63 | 82.20 ± 2.18 | **87.60 ± 0.30** | 71.42 ± 4.43 | 82.78 ±0.30 |
| | Reddit | 62.34 ± 0.54 | 61.60 ± 0.72 | 63.47 ± 0.36 | 64.85 ± 0.85 | 63.67 ± 0.41 | 60.83 ± 0.25 | 64.50 ± 0.26 | 65.37 ± 0.60 | **66.02 ± 0.41** |
| | MOOC | 63.22 ± 1.55 | 62.93 ± 1.24 | 76.73 ± 0.29 | 77.07 ± 3.41 | 74.68 ± 0.68 | 74.27 ± 0.53 | 74.00 ± 0.97 | 80.82 ± 0.30 | **81.63 ± 0.33** |
| | LastFM | 70.39 ± 4.31 | 71.45 ± 1.76 | 76.27 ± 0.25 | 66.65 ± 6.11 | 71.33 ± 0.47 | 65.78 ± 0.65 | 76.42 ± 0.22 | 76.35 ± 0.52 | **77.28 ± 0.21** |
| | Enron | 65.86 ± 3.71 | 62.08 ± 2.27 | 61.40 ± 1.31 | 62.91 ± 1.16 | 60.70 ± 0.36 | 67.11 ± 0.62 | 72.37 ± 1.37 | 67.07 ± 0.62 | **73.01 ± 0.88** |
| | Social Evo. | 88.51 ± 0.87 | 88.72 ± 1.10 | 93.97 ± 0.54 | 90.66 ± 1.62 | 79.83 ± 0.38 | 94.10 ± 0.31 | 94.01 ± 0.47 | **96.82 ± 0.16** | 96.69 ± 0.14 |
| | UCI | 63.11 ± 2.27 | 52.47 ± 2.06 | 70.52 ± 0.93 | 70.78 ± 0.78 | 64.54 ± 0.47 | 76.71 ± 1.00 | **82.66 ± 0.49** | 72.13 ± 1.87 | 82.35 ± 0.39 |
| **Avg.Rank** | | 5.71 | 6.14 | 5.46 | 3.71 | 4.14 | 3.85 | 2.85 | 2.57 | **1.28** |
| ind | Wikipedia | 68.70 ± 0.39 | 62.19 ± 1.28 | 84.17 ± 0.22 | 81.77 ± 0.32 | 67.24 ± 1.63 | 82.20 ± 2.18 | **87.60 ± 0.29** | 71.42 ± 4.43 | 87.54 ± 0.26 |
| | Reddit | 62.32 ± 0.54 | 61.58 ± 0.72 | 63.40 ± 0.36 | 64.84 ± 0.84 | 63.65 ± 0.41 | 60.81 ± 0.26 | 64.49 ± 0.25 | **65.35 ± 0.60** | 64.98 ± 0.20 |
| | MOOC | 63.22 ± 1.55 | 62.92 ± 1.24 | 76.72 ± 0.30 | 77.07 ± 3.40 | 74.69 ± 0.68 | 74.28 ± 0.53 | 73.99 ± 0.97 | 80.82 ± 0.30 | **81.41 ± 0.31** |
| | LastFM | 70.39 ± 4.31 | 71.45 ± 1.75 | 76.28 ± 0.25 | 69.46 ± 4.65 | 71.33 ± 0.47 | 65.78 ± 0.65 | 76.42 ± 0.22 | 76.35 ± 0.52 | **77.01 ± 0.43** |
| | Enron | 65.86 ± 3.71 | 62.08 ± 2.27 | 61.40 ± 1.30 | 62.90 ± 1.16 | 60.72 ± 0.36 | 67.11 ± 0.62 | 72.37 ± 1.38 | 67.07 ± 0.62 | **72.85 ± 0.81** |
| | Social Evo. | 88.51 ± 0.87 | 88.72 ± 1.10 | 93.97 ± 0.54 | 90.65 ± 1.62 | 79.83 ± 0.39 | 94.10 ± 0.32 | 94.01 ± 0.47 | **96.82 ± 0.17** | 96.91 ± 0.12 |
| | UCI | 63.16 ± 2.27 | 52.47 ± 2.09 | 70.49 ± 0.93 | 70.73 ± 0.79 | 64.54 ± 0.47 | 76.65 ± 0.99 | 81.64 ± 0.49 | 72.13 ± 1.86 | **82.06 ± 0.58** |
| **Avg.Rank** | | 6.62 | 7.14 | 4.15 | 4.14 | 5.71 | 4.57 | 3.57 | 2.43 | **1.28** |

Table 2: AP for inductive link prediction with three different negative sampling strategies.

transductive setting and thus in Table 2 and 6 we omit this method. We find that our FreeDyG out-performs other baselines in most scenarios and the average ranking of FreeDyG is close to 1, which is far superior to the second one, i.e., DyGFormer. In addition, it's notable that DyGFormer, when adopting the "hist" negative sampling strategy, experiences a much more significant performance decline compared to the "rnd" strategy. This underscores a limitation in DyGFormer's neighbor encoding approach, which simply relies on the historical co-occurrence counts of neighbors to determine the likelihood of node interactions. In contrast, our method takes into account the historical frequency patterns of node interactions, leading to improved performance.

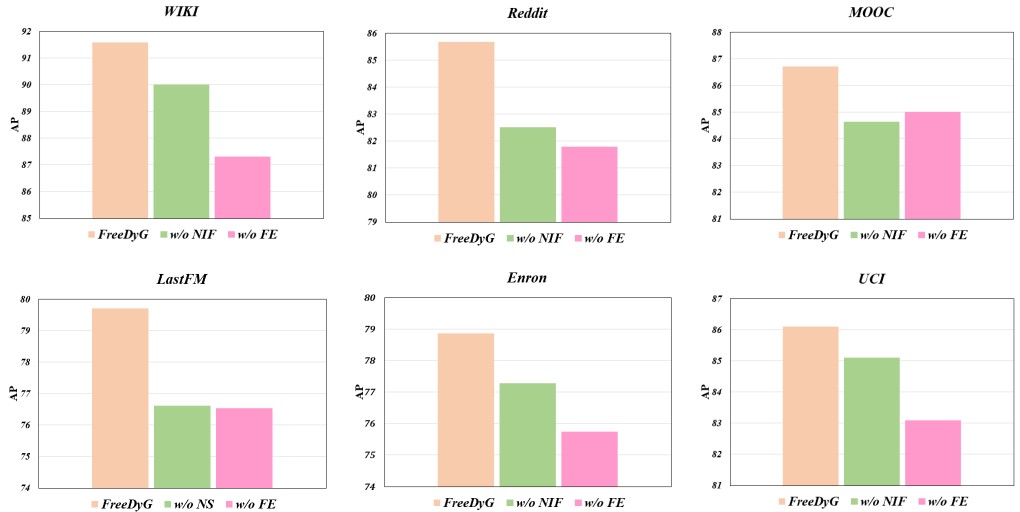

Figure 3: Ablation study of FreeDyG under historical negative sampling setting, where w/o NS and w/o FE represent FreeDyG without node interaction frequency encoding module and FE layer, respectively.

We also conduct a comparative analysis of performance, training time per epoch (measured in seconds), and the size of trainable parameters (measured in Megabyte, i.e., MB) between FreeDyG and baseline methods on the WIKI and Reddit datasets. The results are shown in Figure 2. It is evident that the RW-based method, i.e., CAWN, not only requires a longer training time but also has a substantial number of parameters. On the other hand, simpler methods such as JODIE may have fewer parameters, but there is a significant performance gap compared to the best-performing method. In contrast, FreeDyG achieves the best performance with a small size of trainable parameters and a moderate training time required per epoch.

### 4.5 ABLATION STUDY

Next, we conduct an ablation study on our FreeDyG model to assess the effectiveness of the proposed node interaction frequency encoding approach and frequency-enhancing layer. We refer to FreeDyG without these two modules as **w/o NIF** and **w/o FE** respectively. Figure 3 illustrates the performance comparison under the historical negative sampling setting. Due to the page limit, the performance comparison under the random negative sampling setting is shown in Figure 5 in the Appendix. Both variants exhibit inferior performance across all datasets, highlighting the efficacy of the modules we proposed. It is noteworthy that under different negative sampling settings, distinct modules exhibit varying degrees of importance. In the historical negative sampling setting, the performance decreases more significantly without FE layer. Whereas, under the random negative sampling setting, the NIF encoding module has a more pronounced impact on performance.

The reason behind this phenomenon is as follows: under the random negative sampling setting, where the target node of a negative edge is sampled from the entire graph, the negative samples are significantly easier to distinguish, and the model tends to predict future interactions between node pairs that have previously interacted. Consequently, the NIF encoding, which captures neighbor interactions, assumes a more critical role. However, in the historical negative sampling setting, where the target node of a negative edge is sampled from the source node's historical neighbors, the NIF encoding may introduce more false positives on negative samples. This necessitates the importance of FE layer in capturing temporal or shifting signals. It is evident that the FE component holds greater significance in this setting.

### 4.6 HYPERPARAMETER STUDY

In this section, we analyze the impact of the number of sampled neighbors ($L$) on the performance of FreeDyG. The results are presented in Table 3. We observe that the performance tends to be suboptimal when only a few neighbors are sampled (e.g., $L = 10$), primarily due to the lack of sufficient information. Furthermore, the optimal number of sampled neighbors varies across different datasets.

| | L=10 | | L=20 | | L=32 | | L=64 | | L=100 | |
|---|---|---|---|---|---|---|---|---|---|---|
| | AP | AUC-ROC | AP | AUC-ROC | AP | AUC-ROC | AP | AUC-ROC | AP | AUC-ROC |
| Wikipedia | 99.15 ± 0.01 | 99.30 ± 0.01 | **99.26 ± 0.01** | **99.41 ± 0.01** | 99.23 ± 0.01 | 99.40 ± 0.01 | 99.14 ± 0.03 | 99.19 ± 0.02 | 99.01 ± 0.04 | 99.10 ± 0.09 |
| Reddit | 99.00 ± 0.02 | 99.06 ± 0.02 | 99.21 ± 0.01 | 99.22 ± 0.01 | 99.37 ± 0.01 | 99.40 ± 0.01 | **99.48 ± 0.01** | **99.50 ± 0.01** | 99.45 ± 0.01 | 99.46 ± 0.01 |
| MOOC | 87.10 ± 0.83 | 87.44 ± 1.01 | 88.01 ± 0.30 | 88.17 ± 0.39 | 89.26 ± 0.20 | 89.37 ± 0.61 | **89.61 ± 0.19** | **89.93 ± 0.35** | 88.57 ± 0.52 | 88.71 ± 0.74 |
| LastFM | 85.71 ± 0.22 | 86.64 ± 0.20 | 87.14 ± 0.20 | 87.82 ± 0.26 | 91.13 ± 0.17 | 90.71 ± 0.20 | 90.71 ± 0.22 | 91.53 ± 0.19 | **92.15 ± 0.16** | **93.42 ± 0.15** |
| Enron | 91.46 ± 0.19 | 92.09 ± 0.33 | **92.51 ± 0.05** | **94.01 ± 0.11** | 92.24 ± 0.08 | 93.71 ± 0.09 | 92.43 ± 0.12 | 94.00 ± 0.20 | 91.93 ± 0.31 | 92.21 ± 0.40 |
| Social Evo. | 94.01 ± 0.04 | 95.74 ± 0.05 | 94.58 ± 0.02 | 96.46 ± 0.4 | **94.91 ± 0.01** | **96.59 ± 0.04** | 94.03 ± 0.07 | 95.70 ± 0.13 | 94.06 ± 0.20 | 94.91 ± 0.18 |
| UCI | 95.58 ± 0.19 | 94.21 ± 0.33 | 96.23 ± 0.11 | 94.97 ± 0.26 | **96.28 ± 0.11** | **95.00 ± 0.21** | 95.93 ± 0.17 | 94.46 ± 0.28 | 95.99 ± 0.27 | 94.52 ± 0.43 |

Table 3: Performance comparison of different sample numbers of historical neighbors.

Specifically, datasets characterized by a higher frequency of the "shift" phenomenon (e.g., MOOC, LastFM) Poursafaei et al. (2022) require a larger number of samples to effectively capture hidden patterns and achieve optimal performance. This arises from the essential requirement of FreeDyG to learn from more nuanced frequency domain features of interaction behaviors over time (i.e., DFT of a longer time sequence comprises a greater number of refined frequency domain components).

## 5 RELATED WORK

**Dynamic graph representation learning.** Existing methods can be roughly grouped into two categories: discrete-time dynamic graph (DTDG) models and continuous-time dynamic graph (CTDG) models. DTDG models typically generate discrete snapshots of graphs and fuse information extracted from different snapshots Pareja et al. (2020); Zhao et al. (2019); Yang et al.. However, they often suffer from information loss, as time discretization can miss out on capturing important interactions. To solve it, there is growing interest in designing CTDG models that treat dynamic graph data as link streams and directly learn node representations from interactions that occur continuously. Specifically, existing CTDG models Xu et al. (2020); Wang et al. (2021c;a) commonly employ Recurrent Neural Networks (RNNs) or Self-Attention mechanisms as their basic mechanism. Building upon this, some methods incorporate additional techniques like memory networks Rossi et al. (2020); Kumar et al. (2019); Trivedi et al. (2019), ordinary differential equations (ODE) Luo et al. (2023); Liang et al. (2022), random walk (RW) Wang et al. (2021d); Jin et al. (2022b), temporal point processes (TPP) Chang et al. (2020); Huang et al. (2020) to better learn the continuous temporal information.

**Fourier Transform.** Fourier Transform is a basic technique in the digital signal processing domain Reddy (2018); Pitas (2000) over the decades. Numerous studies have integrated it into areas like computer vision Huang et al. (2023); Wang et al. (2020) and natural language processing Tamkin et al. (2020); Lee-Thorp et al. (2021). More recent research efforts have sought to harness the power of Fourier transform-enhanced models for tasks such as long-term time series forecasting Wu et al. (2023); Zhou et al. (2022; 2021).

## 6 CONCLUSION

In this paper, we present FreeDyG, a frequency-enhanced continuous-time dynamic graph model designed specifically for link prediction. Our approach includes a novel frequency-enhanced MLP-Mixer layer, which effectively captures periodic temporal patterns and the "shift" phenomenon observed in the frequency domain. We also introduce a node interaction frequency encoder that simultaneously extracts the information of interaction frequency and the proportion of common neighbors between node pairs. Extensive experiments conducted on various real-world datasets demonstrate the effectiveness and efficiency of our proposed FreeDyG model. In future work, we plan to extend our method to handle continuous-time dynamic graphs with both edge insertion and deletion events.

## 7 ACKNOWLEDGEMENT

The research presented in this paper is supported in part by the National Natural Science Foundation of China (62372362).

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

## A  DETAILS OF DATASETS.

| Dataset | Nodes | Edges | Unique Edges | Node/Link Feature | Time Granularity | Duration | density |
|---------|-------|-------|--------------|-------------------|------------------|----------|---------|
| WIKI | 9227 | 157474 | 18257 | 0/172 | Unix timestamp | 1 month | 4.30E-03 |
| REDDIT | 10984 | 672447 | 78516 | 0/172 | Unix timestamp | 1 month | 8.51E-03 |
| MOOC | 7144 | 411749 | 178443 | 0/4 | Unix timestamp | 17 month | 1.26E-02 |
| LastFm | 1980 | 1293103 | 154993 | 0/0 | Unix timestamp | 1 month | 5.57E-01 |
| Enron | 184 | 125235 | 3125 | 0/0 | Unix timestamp | 3 years | 5.53E+00 |
| Social Evo. | 74 | 2099519 | 4486 | 0/2 | Unix timestamp | 8 months | 5.36E+02 |
| UCI | 1899 | 59835 | 20296 | 0/0 | Unix timestamp | 196 days | 3.66E-02 |

Table 4: Dataset statistics

**Wiki**[1]: A dataset tracking user edits on Wikipedia pages, is represented as a bipartite interaction graph that contains interactions(edits) between users and Wikipedia pages. Nodes represent users and pages, and links denote the editing behaviors with timestamps. Each interaction is associated with a 172-dimensional Linguistic Inquiry and Word Count (LIWC) feature. This dataset additionally contains dynamic labels that indicate whether users are temporarily banned from editing.

**Reddit**[2]: A dataset tracking users posting in Reddit, is bipartite and records the posts of users under subreddits for one month. Users and subreddits are the nodes, and links are the timestamped posting requests. Each link has a 172-dimensional LIWC feature. This dataset also includes dynamic labels representing whether users are banned from posting.

**MOOC**[3]: is a bipartite interaction network of online sources, where nodes are students and course content units (e.g., videos and problem sets). Each link denotes a student's access behavior to a specific content unit and is assigned a 4-dimensional feature.

**LastFM**[4]: is bipartite and consists of the information about which songs were listened to by which users over one month. Users and songs are nodes, and links denote the listening behaviors of users.

**Enron**[5]: is an email correspondence dataset containing around 50K emails exchanged among employees of the ENRON energy company over a three-year period. This dataset has no attributes.

**Social Evo.**[6]: is a mobile phone proximity network that monitors the daily activities of an entire undergraduate dormitory for a period of eight months, where each link has a 2-dimensional feature.

**UCI**[7]: is a Facebook-like, unattributed online communication network among students of the University of California at Irvine, along with timestamps with the temporal granularity of seconds.

## B  DETAIL DESCRIPTIONS OF BASELINES

**JODIE** is an RNN-based method. Denote $x_i(t)$ as the embedding of node $v_i$ at timestamp $t$, $x_{ij}^{link}(t)$ as the link feature between $v_i$, $v_j$ at timestamp $t$, and $m_i$ as the timestamp that $v_i$ latest interact with other node. When an interaction between $v_i$, $v_j$ happens at timestamp $t$, JODIE updates the temporal embedding using RNN by $x_i(t) = RNN(x_i(m_i), x_j(m_j), x_{ij}^{link}(t), t - m_i)$, Then, the embedding of node $v_i$ at timestamp $t_0$ is computed by $h_i(t_0) = (1 + (t_0 - m_i)w) \cdot x_i(m_i)$.

**TGAT** is a self-attention-based method that could capture spatial and temporal information simultaneously. TGAT first concatenates the raw feature $x_i$ with a trainable time encoding $z(t)$, i.e., $x_i(t) = [x_i || z(t)]$ and $z(t) = cos(tw + b)$. Then, self-attention is applied to produce node representation $hi(t_0) = SAM(x_i(t_0), x_u(h_u)|u \in N_{t_0}(i))$, where $N_{t_0}(i)$ denotes the neighbors of node $v_i$

---

[1]Download from http://snap.stanford.edu/jodie/wikipedia.csv

[2]Download from http://snap.stanford.edu/jodie/reddit.csv

[3]Download from http://snap.stanford.edu/jodie/mooc.csv

[4]Download from http://snap.stanford.edu/jodie/lastfm.csv

[5]Download from https://zenodo.org/record/7213796#.Y1cO6y8r30o

[6]Download from https://zenodo.org/record/7213796#.Y1cO6y8r30o

[7]Download from https://zenodo.org/record/7213796#.Y1cO6y8r30o

at time $t_0$ and $h_u$ denotes the timestamp of the latest interaction of node $u$. Finally, the prediction on any node pair at time $t_0$ is computed by $MLP([h_i(t_0)||h_j(t_0)])$.

**TGN** is a mixture of RNN- and self-attention-based method. TGN utilizes a memory module to store and update the (memory) state $s_i(t)$ of node $i$. The state of node $i$ is expected to represent $i$'s history in a compressed format. Given the memory updater as $mem$, when an link $e_{ij}(t)$ connecting node $i$ is observed, node $i$'s state is updated as $s_i(t) = mem(s_i(t^-), s_j(t^-)||e_{ij}(t))$. where $s_i(t^-)$ is the memory state of node $i$ just before time $t$. $||$ is the concatenation operator, and node $j$ is $i$'s neighbor connected by $e_{ij}(t)$. The implementation of $mem$ is a recurrent neural network (RNN), and node $i$'s embedding is computed by aggregating information from its K-hop temporal neighborhood using self-attention.

**DyRep** is an RNN-based method that updates node states upon each interaction. It also includes a temporal-attentive aggregation module to consider the temporally evolving structural information in dynamic graphs.

**DyGFormer** is a self-attention based method. Specifically, for node $n_i$, DyGFormer just retrieves the features of involved neighbors and links based on the given features to represent their encodings. Instead of learning at the interaction level, DyGFormer splits each source/destination node's sequence into multiple patches and then feeds them to transformer (Vaswani et al., 2017).

**GraphMixer** is a simple MLP-based architecture, that uses a fixed time encoding function rather than the trainable version and incorporates it into a link encoder based on MLP-Mixer to learn from temporal links. A node encoder with neighbor mean-pooing is employed to summarize node features.

**TCL** is a self-attention based method. It first generates each node's interaction sequence by performing a breadth-first search algorithm on the temporal dependency interaction sub-graph. Then, it presents a graph transformer that considers both graph topology and temporal information to learn node representations. It also incorporates a cross-attention operation for modeling the interdependencies of two interaction nodes.

**CAWN** is a mixer of RNN- and self-attention-based method that proposes to represent network dynamics by extracting temporal network motifs using temporal random walks (CAWs). CAWs replace node identities with the hitting counts of the nodes based on a set of sampled walks to establish the correlation between motifs. Then, the extracted motifs are fed into RNNs to encode each walk as a representation and use self-attention to aggregate the representations of multi-walks into a single vector for downstream tasks.

**EdgeBank** is a pure memory-based approach without trainable parameters for transductive dynamic link prediction. It stores the observed interactions in the memory unit and updates the memory through various strategies. An interaction will be predicted as positive if it is retained in the memory, and negative otherwise.

## C  EVALUATION DETAILS OF SAMPLING STRATEGIES

In the evaluation stage, we combine the original test set, treated as positive samples, with additional negative samples. Specifically, the ratio of positive samples to negative samples is set to 1:1. We employ three different negative sampling strategies, as described in Poursafaei et al. (2022): 1) **Random Negative Sampling Strategy**: This strategy randomly samples negative edges from nearly all possible node pairs within the graphs. 2) **Historical Negative Sampling Strategy**: Here, negative edges are sampled from the set of edges observed in previous timestamps but are absent in the current step. 3) **Inductive Negative Sampling Strategy**: Negative edges are sampled from unseen edges that were not encountered during the training phase.

| NSS | Datasets | JODIE | DyRep | TGAT | TGN | CAWN | EdgeBank | TCL | GraphMixer | DyGFormer | FreeDyG |
|---|---|---|---|---|---|---|---|---|---|---|---|
| rnd | Wikipedia | 96.33 ± 0.07 | 94.37 ± 0.09 | 96.67 ± 0.07 | 98.37 ± 0.07 | 98.54 ± 0.04 | 90.78 ± 0.00 | 95.84 ± 0.18 | 96.92 ± 0.03 | 98.91 ± 0.02 | **99.41 ± 0.01** |
| | Reddit | 98.31 ± 0.05 | 98.17 ± 0.05 | 98.47 ± 0.02 | 98.60 ± 0.06 | 99.01 ± 0.01 | 95.37 ± 0.00 | 97.42 ± 0.02 | 97.17 ± 0.02 | 99.15 ± 0.01 | **99.50 ± 0.01** |
| | MOOC | 83.81 ± 2.09 | 85.03 ± 0.58 | 87.11 ± 0.19 | **91.21 ± 1.15** | 80.38 ± 0.26 | 60.86 ± 0.00 | 83.12 ± 0.18 | 84.01 ± 0.17 | 87.91 ± 0.58 | 89.93 ± 0.35 |
| | LastFM | 70.49 ± 1.66 | 71.16 ± 1.89 | 71.59 ± 0.18 | 78.47 ± 2.94 | 85.92 ± 0.10 | 83.77 ± 0.00 | 64.06 ± 1.16 | 73.53 ± 0.12 | 93.05 ± 0.10 | **93.42 ± 0.15** |
| | Enron | 87.96 ± 0.52 | 84.89 ± 3.00 | 68.89 ± 1.10 | 88.32 ± 0.99 | 90.45 ± 0.14 | 87.05 ± 0.00 | 75.74 ± 0.72 | 84.38 ± 0.21 | 93.33 ± 0.13 | **94.01 ± 0.11** |
| | Social Evo. | 92.05 ± 0.46 | 90.76 ± 0.21 | 94.76 ± 0.16 | 95.39 ± 0.17 | 87.34 ± 0.08 | 81.60 ± 0.00 | 94.84 ± 0.17 | 95.23 ± 0.07 | 96.30 ± 0.01 | **96.59 ± 0.04** |
| | UCI | 90.44 ± 0.49 | 68.77 ± 2.34 | 78.53 ± 0.74 | 92.03 ± 1.13 | 93.87 ± 0.08 | 77.30 ± 0.00 | 87.82 ± 1.36 | 91.81 ± 0.67 | 94.49 ± 0.26 | **95.00 ± 0.21** |
| **Avg.Rank** | | 7.14 | 8.57 | 6.14 | 3.31 | 3.78 | 4.86 | 6.14 | 4.78 | 2.14 | **1.14** |
| hist | Wikipedia | 80.77 ± 0.73 | 77.74 ± 0.33 | 82.87 ± 0.22 | 82.74 ± 0.32 | 67.84 ± 0.64 | 77.27 ± 0.00 | 85.76 ± 0.46 | **87.68 ± 0.17** | 78.80 ± 1.95 | 82.78 ± 0.30 |
| | Reddit | 80.52 ± 0.32 | 80.15 ± 0.18 | 79.33 ± 0.16 | 81.11 ± 0.19 | 80.27 ± 0.30 | 78.58 ± 0.00 | 76.49 ± 0.16 | 77.80 ± 0.12 | 80.54 ± 0.29 | **85.92 ± 0.10** |
| | MOOC | 82.75 ± 0.83 | 81.06 ± 0.94 | 80.81 ± 0.67 | 88.00 ± 1.80 | 71.57 ± 1.07 | 61.90 ± 0.00 | 72.09 ± 0.56 | 76.68 ± 1.40 | 87.04 ± 0.35 | **88.32 ± 0.99** |
| | LastFM | 75.22 ± 2.36 | 74.65 ± 1.98 | 64.27 ± 0.26 | 77.97 ± 3.04 | 68.88 ± 0.24 | 78.09 ± 0.00 | 47.24 ± 3.13 | 64.21 ± 0.73 | **78.78 ± 0.35** | 73.53 ± 0.12 |
| | Enron | 75.39 ± 2.37 | 74.69 ± 3.55 | 61.85 ± 1.43 | 77.09 ± 2.22 | 65.10 ± 0.34 | **79.59 ± 0.00** | 67.95 ± 0.88 | 75.27 ± 1.14 | 76.55 ± 0.52 | 75.74 ± 0.72 |
| | Social Evo. | 90.06 ± 3.15 | 93.12 ± 0.34 | 93.08 ± 0.59 | 94.71 ± 0.53 | 87.43 ± 0.15 | 85.81 ± 0.00 | 93.44 ± 0.68 | 94.39 ± 0.31 | 97.28 ± 0.07 | **97.42 ± 0.02** |
| | UCI | 78.64 ± 3.50 | 57.91 ± 3.12 | 58.89 ± 1.57 | 77.25 ± 2.68 | 57.86 ± 0.15 | 69.56 ± 0.00 | 72.25 ± 3.46 | 77.54 ± 2.02 | 76.97 ± 0.24 | **80.38 ± 0.26** |
| **Avg.Rank** | | 4.78 | 5.85 | 6.01 | 3.85 | 6.54 | 7.14 | 4.85 | 4.14 | 2.85 | **2.14** |
| ind | Wikipedia | 70.96 ± 0.78 | 67.36 ± 0.96 | 81.93 ± 0.22 | 80.97 ± 0.31 | 70.95 ± 0.95 | 81.73 ± 0.00 | 82.19 ± 0.48 | **84.28 ± 0.30** | 75.09 ± 3.70 | 82.74 ± 0.32 |
| | Reddit | 83.51 ± 0.15 | 82.90 ± 0.31 | 87.13 ± 0.20 | 84.56 ± 0.24 | **88.04 ± 0.29** | 85.93 ± 0.00 | 84.67 ± 0.29 | 82.21 ± 0.13 | 86.23 ± 0.51 | 84.38 ± 0.21 |
| | MOOC | 66.63 ± 2.30 | 63.26 ± 1.01 | 73.18 ± 0.33 | 77.44 ± 2.86 | 70.32 ± 1.43 | 48.18 ± 0.00 | 70.36 ± 0.37 | 72.45 ± 0.72 | **80.76 ± 0.76** | 78.47 ± 0.94 |
| | LastFM | 61.32 ± 3.49 | 62.15 ± 2.12 | 63.99 ± 0.21 | 65.46 ± 4.27 | 67.92 ± 0.44 | **77.37 ± 0.00** | 46.93 ± 2.59 | 60.22 ± 0.32 | 69.25 ± 0.36 | 72.30 ± 0.59 |
| | Enron | 70.92 ± 1.05 | 68.73 ± 1.34 | 60.45 ± 2.12 | 71.34 ± 2.46 | 75.17 ± 0.50 | 75.00 ± 0.00 | 67.64 ± 0.86 | 71.53 ± 0.85 | 74.07 ± 0.64 | **77.27 ± 0.61** |
| | Social Evo. | 90.01 ± 3.19 | 93.07 ± 0.38 | 92.94 ± 0.61 | 95.24 ± 0.56 | 89.93 ± 0.15 | 87.88 ± 0.00 | 93.44 ± 0.72 | 94.22 ± 0.32 | 97.51 ± 0.06 | **98.47 ± 0.02** |
| | UCI | 64.14 ± 1.26 | 54.25 ± 2.01 | 60.80 ± 1.01 | 64.11 ± 1.04 | 58.06 ± 0.26 | 58.03 ± 0.00 | 70.05 ± 1.86 | 74.59 ± 0.74 | 65.96 ± 1.18 | **75.39 ± 0.57** |
| **Avg.Rank** | | 6.0 | 5.77 | 4.77 | 3.46 | 4.0 | 5.35 | 5.46 | 4.57 | 2.85 | **2.14** |

Table 5: AUC-ROC for transductive dynamic link prediction with different sampling strategies.

| NSS | Datasets | JODIE | DyRep | TGAT | TGN | CAWN | TCL | GraphMixer | DyGFormer | FreeDyG |
|---|---|---|---|---|---|---|---|---|---|---|
| rnd | Wikipedia | 94.33 ± 0.27 | 91.49 ± 0.45 | 95.90 ± 0.09 | 97.72 ± 0.03 | 98.03 ± 0.04 | 95.57 ± 0.20 | 96.30 ± 0.04 | 98.48 ± 0.03 | **99.01 ± 0.02** |
| | Reddit | 96.52 ± 0.13 | 96.05 ± 0.12 | 96.98 ± 0.04 | 97.39 ± 0.07 | 98.42 ± 0.02 | 93.80 ± 0.07 | 94.97 ± 0.05 | 98.71 ± 0.01 | **98.84 ± 0.01** |
| | MOOC | 83.16 ± 1.30 | 84.03 ± 0.49 | 86.84 ± 0.17 | **91.24 ± 0.99** | 81.86 ± 0.25 | 81.43 ± 0.19 | 82.77 ± 0.24 | 87.62 ± 0.51 | 87.01 ± 0.74 |
| | LastFM | 81.13 ± 3.39 | 82.24 ± 1.51 | 76.99 ± 0.29 | 82.61 ± 3.15 | 87.82 ± 0.12 | 70.84 ± 0.85 | 80.37 ± 0.18 | 94.08 ± 0.08 | **94.32 ± 0.03** |
| | Enron | 81.96 ± 1.34 | 76.34 ± 4.20 | 64.63 ± 1.74 | 78.83 ± 1.11 | 87.02 ± 0.50 | 72.33 ± 0.99 | 76.51 ± 0.71 | **90.69 ± 0.26** | 89.51 ± 0.20 |
| | Social Evo. | 93.70 ± 0.29 | 91.18 ± 0.49 | 93.41 ± 0.19 | 93.43 ± 0.59 | 84.73 ± 0.27 | 93.71 ± 0.18 | 94.09 ± 0.07 | 95.29 ± 0.03 | **96.41 ± 0.07** |
| | UCI | 78.80 ± 0.94 | 58.08 ± 1.81 | 77.64 ± 0.38 | 86.68 ± 2.29 | 90.40 ± 0.11 | 84.49 ± 1.82 | 89.30 ± 0.57 | 92.63 ± 0.13 | **93.01 ± 0.08** |
| **Avg.Rank** | | 4.69 | 5.85 | 5.31 | 2.85 | 3.38 | 5.31 | 6.0 | 1.86 | **1.43** |
| hist | Wikipedia | 61.86 ± 0.53 | 57.54 ± 1.09 | 78.38 ± 0.20 | 75.75 ± 0.29 | 62.04 ± 0.65 | 79.79 ± 0.96 | **82.87 ± 0.21** | 70.33 ± 0.25 | 82.08 ± 0.32 |
| | Reddit | 61.69 ± 0.39 | 60.45 ± 0.37 | 64.43 ± 0.27 | 64.55 ± 0.50 | 64.94 ± 0.21 | 61.43 ± 0.26 | 64.27 ± 0.13 | 66.08 ± 0.34 | **66.79 ± 0.31** |
| | MOOC | 64.48 ± 1.64 | 64.23 ± 1.29 | 74.08 ± 0.27 | 77.69 ± 3.55 | 71.68 ± 0.94 | 69.82 ± 0.32 | 72.53 ± 0.84 | 80.77 ± 0.63 | **81.52 ± 0.37** |
| | LastFM | 68.44 ± 3.26 | 68.79 ± 1.08 | 69.89 ± 0.28 | 66.99 ± 5.62 | 67.69 ± 0.24 | 55.88 ± 1.85 | 70.07 ± 0.20 | 70.73 ± 0.37 | **72.63 ± 0.16** |
| | Enron | 65.32 ± 3.57 | 61.50 ± 2.50 | 57.84 ± 2.18 | 62.68 ± 1.09 | 62.25 ± 0.40 | 64.06 ± 1.02 | 68.20 ± 1.62 | 65.78 ± 0.42 | **70.09 ± 0.65** |
| | Social Evo. | 88.53 ± 0.55 | 87.93 ± 1.05 | 91.87 ± 0.72 | 92.10 ± 1.22 | 83.54 ± 0.24 | 93.28 ± 0.60 | 93.62 ± 0.35 | 96.91 ± 0.09 | **96.94 ± 0.17** |
| | UCI | 60.24 ± 1.94 | 51.25 ± 2.37 | 62.32 ± 1.18 | 62.69 ± 0.90 | 56.39 ± 0.10 | 70.46 ± 1.94 | 75.98 ± 0.84 | 65.55 ± 1.01 | **76.01 ± 0.75** |
| **Avg.Rank** | | 5.08 | 6.00 | 4.23 | 4.38 | 5.38 | 3.69 | 3.08 | 2.85 | **1.14** |
| ind | Wikipedia | 61.87 ± 0.53 | 57.54 ± 1.09 | 78.38 ± 0.20 | 0 75.76 ± 0.29 | 62.02 ± 0.65 | 79.79 ± 0.96 | 82.88 ± 0.21 | 68.33 ± 2.82 | **83.17 ± 0.31** |
| | Reddit | 61.69 ± 0.39 | 60.44 ± 0.37 | 64.39 ± 0.27 | 64.55 ± 0.50 | **64.91 ± 0.21** | 61.36 ± 0.26 | 64.27 ± 0.13 | 64.80 ± 0.25 | 64.51 ± 0.19 |
| | MOOC | 64.48 ± 1.64 | 64.22 ± 1.29 | 74.07 ± 0.27 | 77.68 ± 3.55 | 71.69 ± 0.94 | 69.83 ± 0.32 | 72.52 ± 0.84 | **80.77 ± 0.63** | 75.81 ± 0.69 |
| | LastFM | 68.44 ± 3.26 | 68.79 ± 1.08 | 69.89 ± 0.28 | 66.99 ± 5.61 | 67.68 ± 0.24 | 55.88 ± 1.85 | 70.07 ± 0.20 | 70.73 ± 0.37 | **71.42 ± 0.33** |
| | Enron | 65.32 ± 3.57 | 61.50 ± 2.50 | 57.83 ± 2.18 | 62.68 ± 1.09 | 62.27 ± 0.40 | 64.05 ± 1.02 | 68.19 ± 1.63 | 65.79 ± 0.42 | **68.79 ± 0.91** |
| | Social Evo. | 88.53 ± 0.55 | 87.93 ± 1.05 | 91.88 ± 0.72 | 92.10 ± 1.22 | 83.54 ± 0.24 | 93.28 ± 0.60 | 93.62 ± 0.35 | **96.91 ± 0.09** | 96.79 ± 0.17 |
| | UCI | 60.27 ± 1.94 | 51.26 ± 2.40 | 62.29 ± 1.17 | 62.66 ± 0.91 | 56.39 ± 0.11 | 70.42 ± 1.93 | **75.97 ± 0.85** | 65.58 ± 1.00 | 73.41 ± 0.88 |
| **Avg.Rank** | | 4.69 | 5.85 | 5.31 | 2.85 | 3.38 | 5.31 | 6.0 | 1.86 | **1.43** |

Table 6: AUC-ROC for inductive dynamic link prediction with different sampling strategies.

# D SUPPLEMENTARY MATERIALS

## D.1 THE PSEUDO-CODE OF FREEDYG

In Algorithm 1, we show the pseudo-code of the training process of FreeDyG. In addition, following the suggestion of the reviewers, we briefly describe the procedure of FFT in Algorithm 2.

---

**Algorithm 1** Training pipeline for FreeDyG

---

**Input:** CTDG $G$, a node pair $(u, v)$ with a specific timestamp $(t)$, the neighbor sample number $L$, maximum training epoch of 200, early stopping strategy with $patience = 20$.

**Output:** The probability of the node pair interacting at timestamp $t$

1: initial $patience = 0$
2: **for** training $epoch = 1, 2, 3...$ **do**
3:     Acquire the $L$ most recent first-hop interaction neighbors of nodes $u$ and $v$ from $G$ prior to timestamp $t$ as $S_u^t$ and $S_v^t$;
4:     **for** $S_u^t$ and $S_v^t$ in parallel **do**
5:         Obtain node encoding $\boldsymbol{Z}_{*,V}^t$ and edge encoding $\boldsymbol{Z}_{*,E}^t$;
6:         Obtain time encoding $\boldsymbol{Z}_{*,T}^t$ as Cong et al. (2023);
7:         Obtain NIF encoding $\boldsymbol{Z}_{*,F}^t$ from Equation 3;
8:         $\boldsymbol{Z}_*^t \leftarrow Z_{*,V}^t + Z_{*,E}^t + Z_{*,T}^t + Z_{*,F}^t$;
9:         **for** Frequency-ehanced MLP-Mixer Layer **do**
10:             $\mathcal{Z}_*^l \leftarrow \mathcal{F}(Z_*^t)$ with Equation 1;
11:             $\widehat{\mathcal{Z}}_*^l \leftarrow W \cdot \mathcal{Z}_*^l$;
12:             $\widetilde{Z}_*^l \leftarrow \mathcal{F}^{-1}\left(\widehat{\mathcal{Z}}_*^l\right)$ with Equation 2;
13:             $Z_*^l \leftarrow \text{LayerNorm}\left(Z_*^l + \text{Dropout}\left(\widetilde{Z}_*^l\right)\right)$
14:             Feed $Z_*^l$ into MLP-Mixer Layer with Equation 10;
15:         **end for**
16:         Weighted aggregation with Equation 11;
17:     **end for**
18:     Conduct link prediction with $\hat{y} \leftarrow Softmax(MLP(RELU(MLP(h_u^t || h_v^t))))$
19:     Loss $\mathcal{L}_p \leftarrow -\frac{1}{K} \sum_{i=1}^{S}(y_i * log\hat{y_i} + (1 - y_i) * log(1 - \hat{y_i}))$
20:     **if** current epoch's metrics worse than the previous epoch's **then**
21:         $patience = patience + 1$
22:     **else**
23:         Save the model parameters from the current epoch
24:     **end if**
25:     **if** $patience = 20$ **then**
26:         Exit training process
27:     **end if**
28: **end for**

---

---

**Algorithm 2** The Pseudo-Code of Fast Fourier Transform

---

1: **Input:** Sequence $X$ of length $L$
2: **Output:** Y {FFT of $X$}
3:
4: **Function** NextPowerOf2($L$) {Return the next power of 2 greater than or equal to $L$}
5: $N \leftarrow 1$ {Initialize $n$ to the smallest power of 2}
6: **while** $N < L$ **do**
7:    $N \leftarrow N \times 2$ {Double $n$ until it is greater than or equal to $L$}
8: **end while**
9: **return** $N$
10:
11: **Function** FFT($X$, $N$)
12: **if** $N = 1$ **then**
13:    **return** $X$
14: **end if**
15: $X_{\text{even}} \leftarrow$ even-index elements of $X[0, \ldots, N-1]$
16: $X_{\text{odd}} \leftarrow$ odd-index elements of $X[0, \ldots, N-1]$ {Divide the sequence into two parts: even-indexed and odd-indexed}
17: $Y_{\text{even}} \leftarrow$ FFT($X_{\text{even}}, N/2$) {Recursively apply FFT to even part}
18: $Y_{\text{odd}} \leftarrow$ FFT($X_{\text{odd}}, N/2$) {Recursively apply FFT to odd part}
19: $Y \leftarrow$ new array of size $N$ {Initialize an array $Y$ of size $N$}
20: **for** $k = 0$ **to** $N/2 - 1$ **do**
21:    $t \leftarrow e^{-2\pi i k/N}$ {Compute the twiddle factor}
22:    $Y[k] \leftarrow Y_{\text{even}}[k] + t \cdot Y_{\text{odd}}[k]$ {Compute the FFT value for index $k$}
23:    $Y[k + N/2] \leftarrow Y_{\text{even}}[k] - t \cdot Y_{\text{odd}}[k]$ {Compute the FFT value for index $k + N/2$}
24: **end for**
25: **return** $Y$
26:
27: **Procedure:**
28: $N \leftarrow$ NextPowerOf2($L$) {Pad the sequence length to the next power of 2}
29: $X_{\text{padded}} \leftarrow X$
30: **while** length of $X_{\text{padded}} < N$ **do**
31:    append 0 to $X_{\text{padded}}$ {Pad the sequence with zeros}
32: **end while**
33: $Y \leftarrow$ FFT($X_{\text{padded}}, N$) {Apply FFT to the padded sequence}

---

### D.2 THE EXPERIMENTAL RESULTS ON MICRO-F1 AND MACRO-F1 METRIC

In Table 7-10, we conduct experiments on all datasets under three negative sampling strategies with both transductive and inductive link prediction. We use Micro-F1 and Macro-F1 scores as the metric. We note that the results are similar to those on AP and AUC-ROC. Specifically, our FreeDyG outperforms other baselines in most scenarios and the average ranking of FreeDyG is close to 1.

| NSS | Datasets | JODIE | DyRep | TGAT | TGN | CAWN | TCL | GraphMixer | DyGFormer | FreeDyG |
|---|---|---|---|---|---|---|---|---|---|---|
| rnd | Wikipedia | 87.47 ± 0.99 | 87.12 ± 0.45 | 89.69 ± 0.15 | 93.09 ± 0.05 | 94.63 ± 0.08 | 88.06 ± 0.50 | 90.57 ± 0.24 | 94.60 ± 0.53 | **95.92 ± 0.09** |
| | reddit | 91.99 ± 0.49 | 93.03 ± 0.21 | 93.90 ± 0.17 | 94.15 ± 0.07 | 95.55 ± 0.03 | 91.68 ± 0.07 | 91.24 ± 0.04 | 95.86 ± 0.20 | **96.03 ± 0.07** |
| | MOOC | 73.02 ± 1.20 | 78.05 ± 1.58 | 78.26 ± 0.38 | **84.34 ± 1.72** | 70.05 ± 0.56 | 74.02 ± 0.32 | 74.76 ± 0.18 | 75.44 ± 1.04 | 76.74 ± 0.97 |
| | LastFM | 59.74 ± 3.96 | 61.01 ± 5.68 | 65.01 ± 0.26 | 54.93 ± 3.44 | 80.94 ± 0.47 | 59.50 ± 0.33 | 66.96 ± 0.44 | 88.13 ± 0.64 | **88.89 ± 0.38** |
| | Enron | 71.74 ± 0.84 | 77.01 ± 2.61 | 63.09 ± 2.39 | 78.36 ± 3.22 | 84.53 ± 0.16 | 70.88 ± 2.79 | 76.47 ± 0.16 | 88.37 ± 0.33 | **88.91 ± 0.20** |
| | Social Evo. | 77.46 ± 4.26 | 83.66 ± 0.62 | 90.42 ± 0.12 | 91.18 ± 0.23 | 80.67 ± 0.31 | 90.56 ± 0.18 | 90.67 ± 0.14 | 93.26 ± 0.10 | **93.87 ± 0.07** |
| | UCI | 67.61 ± 2.97 | 63.49 ± 9.17 | 70.29 ± 1.18 | 79.39 ± 0.67 | 87.84 ± 0.15 | 76.15 ± 0.58 | 83.30 ± 0.99 | 89.34 ± 0.07 | **90.25± 0.08** |
| | Avg.Rank | 7.71 | 6.43 | 5.71 | 4.28 | 4.43 | 7.00 | 5.43 | 2.57 | **1.43** |
| hist | Wikipedia | 65.17 ± 1.27 | 63.35 ± 0.61 | 62.53 ± 0.43 | 59.67 ± 1.26 | 50.18 ± 0.23 | 67.61 ± 0.88 | **69.94 ± 1.44** | 52.87 ± 1.88 | 66.73 ± 1.32 |
| | Reddit | 59.76 ± 0.26 | 61.40 ± 0.76 | 59.27 ± 0.54 | 60.55 ± 0.33 | 54.04 ± 0.03 | 59.22 ± 0.14 | 61.74 ± 0.10 | 49.87 ± 0.14 | **62.74 ± 0.61** |
| | MOOC | 63.74 ± 9.08 | 75.35 ± 0.54 | 72.90 ± 0.37 | **77.77 ± 0.87** | 64.20 ± 1.42 | 59.69 ± 2.06 | 65.58 ± 0.12 | 75.10 ± 5.73 | 76.18 ± 0.87 |
| | LastFM | 49.71 ± 1.53 | 59.42 ± 2.52 | 55.26 ± 0.49 | 50.86 ± 1.06 | 59.85 ± 0.68 | 38.98 ± 2.52 | 56.39 ± 0.62 | **68.59 ± 0.55** | 67.81 ± 0.40 |
| | Enron | 65.91 ± 0.31 | **68.11 ± 2.64** | 56.38 ± 0.64 | 67.63 ± 1.81 | 55.72 ± 0.94 | 66.74 ± 0.34 | 61.48 ± 1.79 | 66.52 ± 1.64 | 66.97 ± 1.28 |
| | Social Evo. | 73.30 ± 0.25 | 86.41 ± 0.38 | 83.47 ± 0.49 | 84.81 ± 1.28 | 86.81 ± 0.17 | 87.70 ± 5.17 | 84.96 ± 0.96 | 89.42 ± 0.69 | **90.02 ± 0.83** |
| | UCI | 62.13 ± 3.38 | 51.82 ± 9.65 | 49.44 ± 0.48 | 56.82 ± 0.70 | 44.78 ± 0.02 | 48.93 ± 0.43 | **78.66 ± 0.02** | 59.96 ± 2.00 | 64.39 ± 1.29 |
| | **Avg.Rank** | 6.14 | 3.86 | 6.57 | 4.71 | 7.00 | 6.00 | 4.00 | 4.71 | **2.00** |
| ind | Wikipedia | 53.90 ± 0.70 | 53.91 ± 0.01 | 59.76 ± 0.28 | 57.25 ± 0.23 | 52.73 ± 0.22 | 63.64 ± 0.17 | **63.78 ± 1.34** | 49.49 ± 0.02 | 58.10 ± 0.74 |
| | Reddit | 58.07 ± 0.12 | 59.52 ± 0.18 | 65.52 ± 0.65 | 58.57 ± 0.41 | 58.76 ± 0.41 | **66.09 ± 0.03** | 63.94 ± 0.19 | 48.65 ± 0.38 | 63.09 ± 0.73 |
| | MOOC | 51.69 ± 3.58 | 57.96 ± 0.32 | 64.54 ± 0.64 | 61.75 ± 2.58 | 62.90 ± 0.91 | 60.11 ± 1.55 | 62.50 ± 0.38 | 67.13 ± 2.26 | **70.06 ± 1.45** |
| | LastFM | 47.55 ± 1.53 | 51.57 ± 2.24 | 55.14 ± 0.66 | 50.34 ± 0.45 | 63.87 ± 0.22 | 39.30 ± 2.32 | 53.42 ± 0.27 | 66.31 ± 0.30 | **66.57 ± 0.71** |
| | Enron | 58.57 ± 0.40 | 61.26 ± 2.27 | 54.75 ± 1.25 | 60.51 ± 0.68 | 64.57 ± 1.18 | **65.66 ± 0.50** | 58.24 ± 0.73 | 56.80 ± 0.74 | 60.09 ± 1.43 |
| | Social Evo. | 73.42 ± 0.40 | 86.49 ± 0.40 | 83.68 ± 0.46 | 86.13 ± 1.02 | 87.72 ± 0.35 | 87.62 ± 5.33 | 84.94 ± 1.05 | 89.40 ± 0.65 | **89.87 ± 0.52** |
| | UCI | 58.34 ± 1.64 | 53.07 ± 2.71 | 51.69 ± 0.25 | 49.75 ± 0.46 | 46.14 ± 0.02 | 50.76 ± 0.84 | **69.43 ± 0.21** | 46.53 ± 0.84 | 60.02 ± 2.61 |
| | **Avg.Rank** | 7.14 | 5.28 | 4.86 | 6.00 | 5.00 | 4.28 | 4.14 | 5.71 | **2.57** |

Table 7: micro-F1 for transductive link prediction with three different negative sampling strategies.

| NSS | Datasets | JODIE | DyRep | TGAT | TGN | CAWN | TCL | GraphMixer | DyGFormer | FreeDyG |
|---|---|---|---|---|---|---|---|---|---|---|
| rnd | Wikipedia | 87.45 ± 0.99 | 87.10 ± 0.45 | 89.69 ± 0.15 | 93.09 ± 0.05 | 94.63 ± 0.08 | 88.06 ± 0.50 | 90.57 ± 0.24 | 94.60 ± 0.53 | **95.92 ± 0.09** |
| | reddit | 91.99 ± 0.49 | 93.03 ± 0.21 | 93.90 ± 0.17 | 94.14 ± 0.07 | 95.55 ± 0.03 | 91.68 ± 0.07 | 91.24 ± 0.04 | 95.86 ± 0.20 | **96.03 ± 0.07** |
| | MOOC | 72.13 ± 1.45 | 77.89 ± 1.69 | 78.23 ± 0.38 | **84.29 ± 1.75** | 69.63 ± 0.67 | 73.91 ± 0.35 | 74.65 ± 0.18 | 75.37 ± 1.09 | 76.63 ± 1.01 |
| | LastFM | 55.61 ± 4.77 | 55.77 ±5.49 | 64.90 ± 0.27 | 42.42 ± 5.57 | 80.86 ± 0.44 | 58.93 ± 0.60 | 66.70 ± 0.44 | 87.95 ± 0.60 | **88.02 ± 0.47** |
| | Enron | 69.51 ± 1.25 | 76.55 ± 2.95 | 62.28 ± 2.65 | 77.75 ± 3.63 | 84.23 ± 0.20 | 69.66 ± 3.13 | 76.04 ± 0.17 | 88.01 ± 0.32 | **88.83 ± 0.21** |
| | Social Evo. | 78.47 ± 3.55 | 83.58 ± 0.66 | 90.41 ± 0.12 | 91.17 ± 0.23 | 80.58± 0.29 | 90.55 ± 0.18 | 90.67 ± 0.14 | 93.26 ± 0.10 | **93.87 ± 0.07** |
| | UCI | 63.43 ± 4.37 | 61.57 ± 11.71 | 70.19 ± 1.18 | 78.88 ± 0.76 | 87.76 ± 0.13 | 76.01 ± 0.62 | 82.87 ± 1.07 | 89.24 ± 0.09 | **90.22 ± 0.08** |
| | Avg.Rank | 8.00 | 6.57 | 5.71 | 4.28 | 4.43 | 6.57 | 5.43 | 2.57 | **1.43** |
| hist | Wikipedia | 62.92 ± 1.80 | 61.21 ± 0.79 | 59.41 ± 0.55 | 55.02 ± 1.99 | 38.90 ± 0.58 | 66.34 ± 1.02 | **68.70 ± 1.74** | 45.72 ± 1.45 | 61.52 ± 1.25 |
| | Reddit | 54.11 ± 0.32 | 56.89 ± 1.19 | 53.73 ± 0.92 | 55.45 ± 0.59 | 44.95± 0.37 | 54.43 ± 0.27 | 58.19 ± 0.19 | 36.58 ± 0.18 | **58.71 ± 0.59** |
| | MOOC | 59.01 ± 14.15 | 77.14 ± 0.50 | 72.73 ± 0.46 | **77.44 ± 0.80** | 63.98 ± 1.37 | 58.61 ± 2.32 | 65.07 ± 0.15 | 74.92 ± 5.87 | 76.03 ± 0.87 |
| | LastFM | 48.17 ± 1.92 | 53.69 ± 2.28 | 54.23 ± 0.78 | 35.63 ± 2.79 | 58.75 ± 0.94 | 35.34 ± 6.15 | 55.68 ± 0.71 | 67.28 ± 0.37 | **67.48 ± 0.42** |
| | Enron | 61.92 ± 0.06 | **66.87 ± 2.99** | 55.35 ± 0.73 | 65.93 ± 2.12 | 51.65 ± 2.12 | 65.93 ± 0.57 | 59.69 ± 1.83 | 64.98 ± 1.91 | 65.33 ± 1.70 |
| | Social Evo. | 71.95 ± 0.07 | 86.37 ± 0.38 | 83.37 ± 0.50 | 84.69 ± 1.35 | 87.75 ± 0.34 | 87.66 ± 5.21 | 84.89 ± 0.98 | 89.23 ± 0.71 | **90.17 ± 0.64** |
| | UCI | 57.97 ± 5.87 | 48.76 ± 12.80 | 46.15 ± 1.10 | 49.27 ± 0.75 | 35.98 ± 0.05 | 42.58 ± 1.60 | **78.43 ± 0.01** | 57.90 ± 2.73 | 65.07 ± 1.15 |
| | **Avg.Rank** | 6.00 | 4.00 | 6.57 | 4.85 | 6.85 | 5.71 | 3.85 | 4.85 | **2.28** |
| ind | Wikipedia | 46.88 ± 1.59 | 47.95 ± 0.05 | 55.48 ± 0.49 | 51.12 ± 0.58 | 42.85 ± 0.57 | **61.48 ± 0.18** | 61.07 ± 1.82 | 36.76 ± 1.03 | 52.71 ± 1.36 |
| | Reddit | 51.34 ± 0.27 | 54.00 ± 0.41 | 62.43 ± 0.94 | 52.29 ± 0.75 | 52.47 ± 0.51 | **63.64 ± 0.07** | 61.04 ± 0.29 | 33.66 ± 0.30 | 61.25 ± 0.81 |
| | MOOC | 40.70 ± 9.95 | 54.12 ± 0.92 | 63.63 ± 0.98 | 58.48 ± 2.91 | 62.73 ± 0.88 | 59.11 ± 1.72 | 61.73 ± 0.45 | 66.74 ± 2.20 | **68.31 ± 1.54** |
| | LastFM | 42.08 ± 2.31 | 40.70 ± 4.93 | 54.07 ± 0.97 | 34.61 ± 1.62 | **63.29 ± 0.89** | 35.78 ± 5.79 | 52.41 ± 0.25 | 51.37 ± 1.04 | 60.42 ± 1.59 |
| | Enron | 50.58 ± 1.45 | 58.24 ± 2.84 | 53.31 ± 1.42 | 56.00 ± 0.59 | 62.66 ± 1.94 | **64.93 ± 0.70** | 55.67 ± 0.51 | 51.37 ± 1.04 | 55.79 ± 1.06 |
| | Social Evo. | 72.07 ± 0.25 | 86.45 ± 0.40 | 83.60 ± 0.48 | 86.05 ± 1.07 | 88.31 ± 0.16 | 87.57 ± 5.38 | 84.88 ± 1.08 | 89.25 ± 0.70 | **89.94 ± 0.47** |
| | UCI | 54.41 ± 4.13 | 48.49 ± 6.79 | 49.11 ± 0.07 | 35.97 ± 0.32 | 37.56 ± 0.05 | 45.18 ± 2.13 | **69.24 ± 0.21** | 38.10 ± 2.10 | 58.47 ± 2.41 |
| | **Avg.Rank** | 7.28 | 5.57 | 4.28 | 6.71 | 4.57 | 3.85 | 4.14 | 6.00 | **2.57** |

Table 8: macro-F1 for transductive link prediction with three different negative sampling strategies.

| NSS | Datasets | JODIE | DyRep | TGAT | TGN | CAWN | TCL | GraphMixer | DyGFormer | FreeDyG |
|---|---|---|---|---|---|---|---|---|---|---|
| rnd | Wikipedia | 84.05 ± 0.89 | 82.60 ± 0.26 | 88.20 ± 0.14 | 91.22 ± 0.27 | 92.86 ± 0.11 | 87.09 ± 0.27 | 89.23 ± 0.09 | _93.13 ± 0.50_ | **94.42 ± 0.05** |
| | reddit | 88.84 ± 0.73 | 88.99 ± 0.48 | 91.06 ± 0.42 | 91.60 ± 0.19 | 94.24 ± 0.04 | 86.39 ± 0.21 | 87.88 ± 0.15 | _94.62 ± 0.22_ | **95.33 ± 0.09** |
| | MOOC | 71.19 ± 1.66 | 76.89 ± 0.66 | _78.00 ± 0.32_ | **83.91 ± 0.76** | 72.01 ± 0.27 | 72.55 ± 0.26 | 73.53 ± 0.17 | 75.24 ± 1.42 | 77.81 ± 1.12 |
| | LastFM | 70.71 ± 4.65 | 70.68 ± 6.38 | 70.10 ± 0.36 | 58.14 ± 6.06 | 83.55 ± 0.08 | 65.86 ± 0.48 | 72.93 ± 0.85 | _90.03 ± 0.25_ | **90.49 ± 0.23** |
| | Enron | 65.50 ± 1.05 | 66.01 ± 3.88 | 59.29 ± 2.13 | 71.54 ± 2.98 | 80.29 ± 0.24 | 68.47 ± 3.03 | 69.50 ± 0.18 | _85.15 ± 0.24_ | **86.34 ± 0.26** |
| | Social Evo. | 84.78 ± 1.22 | 83.54 ± 1.94 | 88.80 ± 0.12 | 88.49 ± 0.53 | 79.42 ± 0.41 | 89.74 ± 0.13 | 89.67 ± 0.08 | _92.16 ± 0.08_ | **92.59 ± 0.07** |
| | UCI | 61.25 ± 2.27 | 56.51 ± 5.09 | 68.41 ± 0.89 | 69.41 ± 2.12 | 85.26 ± 0.55 | 71.13 ± 0.78 | 82.44 ± 0.68 | _88.40 ± 0.07_ | **89.11 ± 0.05** |
| | **Avg.Rank** | 7.43 | 7.00 | 5.85 | 4.85 | 4.57 | 6.43 | 5.14 | 2.43 | **1.28** |
| hist | Wikipedia | 48.81 ± 0.11 | 48.41 ± 0.06 | 56.76 ± 0.38 | 52.89 ± 0.24 | 49.37 ± 0.17 | _61.10 ± 0.25_ | **61.11 ± 1.25** | 48.48 ± 0.27 | 57.43 ± 0.75 |
| | Reddit | 51.50 ± 0.19 | 51.64 ± 0.03 | 54.12 ± 0.11 | 51.67 ± 0.23 | 50.87 ± 0.33 | _54.14 ± 0.27_ | **54.48 ± 0.16** | 48.10 ± 0.38 | 51.07 ± 0.36 |
| | MOOC | 52.98 ± 4.84 | 59.02 ± 0.65 | 65.59 ± 0.25 | 63.13 ± 1.90 | 64.37 ± 1.07 | 59.76 ± 1.56 | 62.65 ± 0.24 | **66.80 ± 1.86** | _66.02 ± 0.75_ |
| | LastFM | 52.09 ± 3.15 | 53.35 ± 4.66 | 57.99 ± 0.61 | 50.37 ± 0.55 | 54.64 ± 0.31 | 42.84 ± 3.28 | 57.88 ± 0.24 | **69.75 ± 0.53** | _68.54 ± 0.48_ |
| | Enron | _60.29 ± 0.83_ | 55.27 ± 1.11 | 52.03 ± 0.60 | 56.84 ± 2.40 | 58.71 ± 0.75 | **63.38 ± 0.57** | 56.05 ± 0.34 | 57.31 ± 0.47 | 58.23 ± 1.07 |
| | Social Evo. | 69.77 ± 0.74 | 71.25 ± 3.58 | 81.07 ± 0.94 | 81.29 ± 3.48 | 88.14 ± 0.73 | 87.69 ± 4.42 | 83.82 ± 0.79 | _89.01 ± 0.53_ | **90.13 ± 0.58** |
| | UCI | 56.71 ± 1.55 | 52.27 ± 0.24 | 52.72 ± 0.09 | 49.16 ± 0.49 | 44.71 ± 0.05 | 50.54 ± 0.89 | **71.02 ± 0.11** | 46.79 ± 0.98 | _62.19 ± 2.09_ |
| | **Avg.Rank** | 6.14 | 7.00 | 4.71 | 5.85 | 5.43 | 4.43 | 3.57 | 4.86 | **3.00** |
| ind | Wikipedia | 48.81 ± 0.11 | 48.41 ± 0.06 | 56.76 ± 0.38 | 52.89 ± 0.24 | 49.37 ± 0.17 | _61.09 ± 0.25_ | **61.10 ± 1.26** | 48.48 ± 0.27 | 57.42 ± 0.75 |
| | Reddit | 51.50 ± 0.19 | 51.64 ± 0.03 | _54.07 ± 0.03_ | 51.67 ± 0.23 | 50.86 ± 0.30 | 54.16 ± 0.24 | **54.48 ± 0.16** | 48.10 ± 0.38 | 51.07 ± 0.36 |
| | MOOC | 52.98 ± 4.85 | 59.02 ± 0.65 | 65.60 ± 0.25 | 63.13 ± 1.89 | 64.37 ± 1.07 | 59.76 ± 1.56 | 62.66 ± 0.24 | **66.80 ± 1.87** | 66.02 ± 0.75 |
| | LastFM | 52.09 ± 3.1 | 53.35 ± 4.66 | 57.99 ± 0.61 | 50.37 ± 0.55 | 54.64 ± 0.31 | 42.83 ± 3.28 | 57.89 ± 0.24 | **69.76 ± 0.53** | _68.54 ± 0.48_ |
| | Enron | _60.29 ± 0.83_ | 55.27 ± 1.11 | 52.04 ± 0.60 | 56.85 ± 2.40 | 58.72 ± 0.76 | **63.39 ± 0.58** | 56.06 ± 0.35 | 57.31 ± 0.47 | 58.23 ± 1.07 |
| | Social Evo. | 69.77 ± 0.74 | 71.25 ± 3.58 | 81.73 ± 0.92 | 81.30 ± 3.47 | 88.20 ± 0.75 | 87.70 ± 4.42 | 83.82 ± 0.79 | _89.01 ± 0.53_ | **90.13 ± 0.58** |
| | UCI | 56.71 ± 1.53 | 52.29 ± 0.25 | 52.62 ± 0.07 | 49.16 ± 0.51 | 44.71 ± 0.05 | 50.58 ± 0.90 | **71.01 ± 0.11** | 46.79 ± 0.99 | _62.19 ± 2.09_ |
| | **Avg.Rank** | 6.14 | 7.00 | 4.57 | 6.00 | 5.43 | 4.43 | 3.57 | 4.85 | **3.00** |

Table 9: micro-F1 for inductive link prediction with three different negative sampling strategies.

| NSS | Datasets | JODIE | DyRep | TGAT | TGN | CAWN | TCL | GraphMixer | DyGFormer | FreeDyG |
|---|---|---|---|---|---|---|---|---|---|---|
| rnd | Wikipedia | 84.02 ± 0.89 | 82.55 ± 0.27 | 88.19 ± 0.14 | 91.22 ± 0.27 | 92.86 ± 0.11 | 87.09 ± 0.27 | 89.23 ± 0.09 | _93.13 ± 0.50_ | **94.42 ± 0.05** |
| | reddit | 88.85 ± 0.72 | 88.99 ± 0.48 | 91.06 ± 0.42 | 91.60 ± 0.19 | 94.24 ± 0.04 | 86.39 ± 0.21 | 87.88 ± 0.15 | _94.62 ± 0.22_ | **95.33 ± 0.09** |
| | MOOC | 70.55 ± 1.47 | 76.80 ± 0.70 | _77.97 ± 0.33_ | **83.87 ± 0.76** | 71.78 ± 0.31 | 72.44 ± 0.28 | 73.43 ± 0.17 | 75.18 ± 1.47 | 77.71 ± 1.14 |
| | LastFM | 66.82 ± 5.98 | 67.76 ± 10.04 | 70.01 ± 0.36 | 47.46 ± 9.02 | 83.42 ± 0.08 | 65.40 ± 0.52 | 72.80 ± 0.87 | _90.14 ± 0.24_ | **90.65 ± 0.25** |
| | Enron | 68.11 ± 0.67 | 64.21 ± 4.73 | 58.61 ± 2.40 | 70.39 ± 3.71 | 79.97 ± 0.30 | 67.34 ± 3.43 | 68.75 ± 0.18 | _84.84 ± 0.24_ | **86.10 ± 0.16** |
| | Social Evo. | 85.03 ± 1.14 | 83.29 ± 2.06 | 88.78 ± 0.12 | 88.43 ± 0.56 | 79.16 ± 0.40 | 89.74 ± 0.13 | 89.67 ± 0.08 | _92.16 ± 0.08_ | **92.59 ± 0.07** |
| | UCI | 55.66 ± 3.90 | 54.23 ± 7.99 | 68.23 ± 0.88 | 67.33 ± 2.91 | 85.21 ± 0.53 | 70.84 ± 0.87 | 82.10 ± 0.75 | _88.31 ± 0.07_ | **89.05 ± 0.04** |
| | Avg.Rank | 7.43 | 7.14 | 5.43 | 5.00 | 4.57 | 6.57 | 5.14 | 2.43 | **1.28** |
| hist | Wikipedia | 40.56 ± 0.65 | 41.01 ± 0.21 | 51.08 ± 0.68 | 44.34 ± 0.78 | 49.37 ± 0.17 | **57.98 ± 0.33** | _57.33 ± 1.96_ | 35.52 ± 0.87 | 50.49 ± 0.59 |
| | Reddit | 42.53 ± 0.57 | 43.77 ± 0.15 | 46.74 ± 0.37 | 41.72 ± 0.54 | 44.95 ± 0.25 | _49.15 ± 0.08_ | 48.60 ± 0.35 | 33.16 ± 0.21 | **49.81 ± 0.76** |
| | MOOC | 44.48 ± 12.08 | 56.69 ± 1.20 | 64.92 ± 0.50 | 60.30 ± 2.21 | 64.31 ± 1.06 | 58.89 ± 1.73 | 62.02 ± 0.30 | **66.39 ± 1.80** | _65.63 ± 1.53_ |
| | LastFM | 49.01 ± 3.41 | 47.68 ± 5.06 | 56.77 ± 0.93 | 34.73 ± 1.83 | 52.05 ± 0.85 | 38.60 ± 6.90 | 56.41 ± 0.22 | **68.61 ± 0.37** | _68.11 ± 0.40_ |
| | Enron | _55.43 ± 1.92_ | 49.32 ± 1.57 | 50.45 ± 0.63 | 52.43 ± 3.86 | 55.24 ± 1.54 | **62.66 ± 0.68** | 53.53 ± 0.11 | 52.70 ± 0.87 | 54.70 ± 0.82 |
| | Social Evo. | 67.53 ± 1.25 | 69.55 ± 4.61 | 80.90 ± 0.97 | 80.88 ± 3.84 | 88.09 ± 0.57 | 87.65 ± 4.46 | 83.73 ± 0.81 | _88.92 ± 0.46_ | **89.28 ± 0.51** |
| | UCI | _50.77 ± 4.58_ | 47.47 ± 5.98 | 50.45 ± 0.38 | 35.17 ± 0.01 | 35.48 ± 0.07 | 45.52 ± 2.05 | **70.83 ± 0.10** | 38.54 ± 2.33 | 50.05 ± 3.15 |
| | **Avg.Rank** | 6.14 | 7.14 | 4.29 | 7.43 | 4.71 | 4.14 | 3.57 | 5.00 | **2.57** |
| ind | Wikipedia | 40.55 ± 0.64 | 41.01 ± 0.21 | 51.08 ± 0.69 | 44.34 ± 0.77 | 37.36 ± 0.41 | **57.97 ± 0.34** | _57.32 ± 1.97_ | 35.52 ± 0.87 | 50.49 ± 0.59 |
| | Reddit | 42.53 ± 0.57 | 43.77 ± 0.15 | 46.65 ± 0.27 | 41.72 ± 0.54 | 39.42 ± 0.38 | _49.14 ± 0.04_ | 48.58 ± 0.35 | 33.16 ± 0.21 | **49.81 ± 0.76** |
| | MOOC | 44.48 ± 12.08 | 56.69 ± 1.20 | 64.92 ± 0.50 | 60.30 ± 2.20 | 64.31 ± 1.06 | 58.90 ± 1.73 | 62.03 ± 0.30 | **66.39 ± 1.80** | _65.63 ± 1.53_ |
| | LastFM | 49.01 ± 3.41 | 47.68 ± 5.06 | 56.78 ± 0.93 | 34.73 ± 1.83 | 52.05 ± 0.85 | 38.60 ± 6.90 | 56.41 ± 0.22 | **68.63 ± 0.37** | _68.11 ± 0.40_ |
| | Enron | _55.43 ± 1.92_ | 49.32 ± 1.57 | 50.45 ± 0.62 | 52.44 ± 3.86 | 55.25 ± 1.55 | **62.67 ± 0.69** | 53.54 ± 0.12 | 52.70 ± 0.87 | 54.70 ± 0.82 |
| | Social Evo. | 67.53 ± 1.25 | 69.55 ± 4.61 | 81.58 ± 0.92 | 80.88 ± 3.83 | 88.12 ± 0.51 | 87.66 ± 4.46 | 83.72 ± 0.81 | _88.92 ± 0.46_ | **89.28 ± 0.51** |
| | UCI | _50.76 ± 4.56_ | 47.50 ± 6.01 | 50.32 ± 0.37 | 35.17 ± 0.02 | 35.49 ± 0.06 | 45.57 ± 2.05 | **70.82 ± 0.11** | 38.56 ± 2.35 | 50.05 ± 3.15 |
| | **Avg.Rank** | 5.86 | 6.86 | 4.29 | 7.14 | 5.57 | 4.14 | 3.57 | 5.00 | **2.57** |

Table 10: macro-F1 for inductive link prediction with three different negative sampling strategies.

## D.3 HYPERPARAMETER STUDY

To enhance the presentation of the results in Table 3, we employ Figure 4 to visually depict the trends observed across varying numbers of sampled historical neighbors.

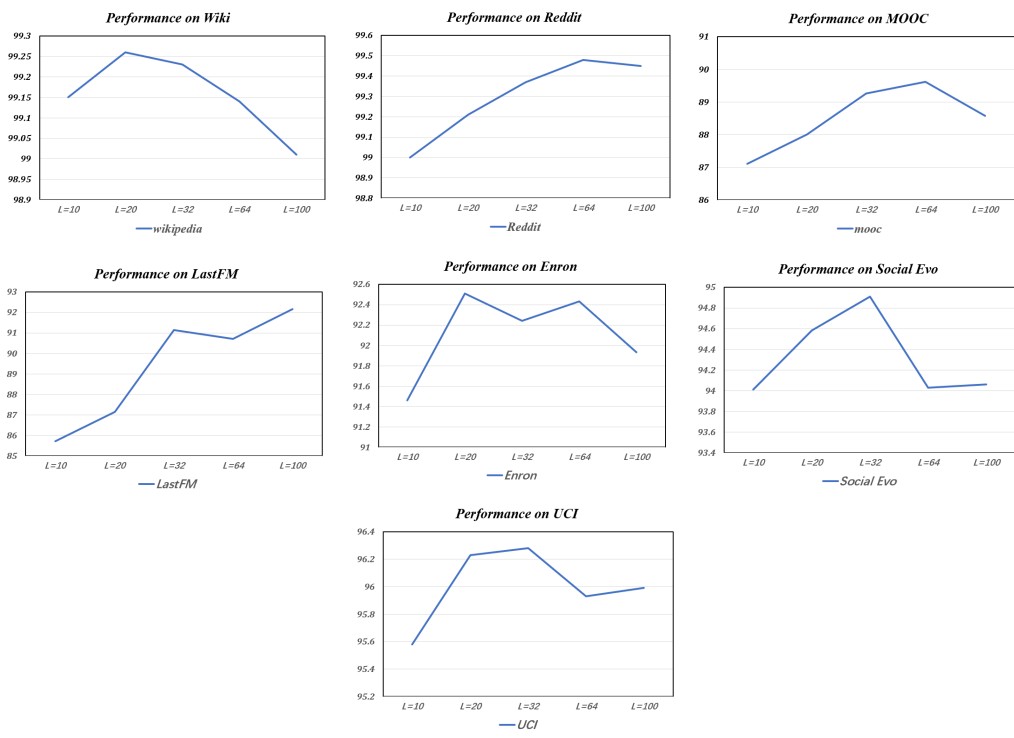

Figure 4: Performance comparison on AP of different number of sampled historical neighbors

## D.4 ABLATION STUDY OF FREEDYG UNDER RANDOM NEGATIVE SAMPLING SETTING

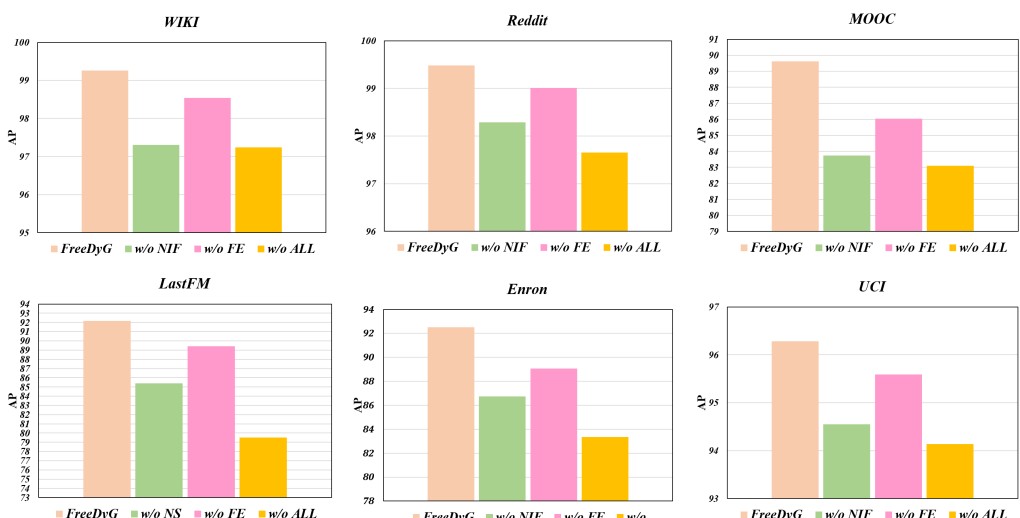

Figure 5: Ablation study of FreeDyG under random negative sampling setting, where w/o NS and w/o FE represent our FreeDyG without node interaction frequency encoding module and FE layer respectively.

