# OpenReview forum: "FreeDyG: Frequency Enhanced Continuous-Time Dynamic Graph Model for Link Prediction"
_ICLR.cc/2024/Conference — ICLR 2024 poster_

### Official Review · Reviewer_VgR1 · 2023-10-28

**Soundness:** 3 good
**Presentation:** 3 good
**Contribution:** 2 fair
**Rating:** 5
**Confidence:** 3

**Summary:**

In this paper, the authors considered the temporal link prediction task on continuous-time dynamic graph (CTDG) and proposed a novel FreeDyG method, which to my knowledge, is the first work to use a frequency-based discrete Fourier transform (DFT) to capture the evolving patterns of CTDG. The overall presentation of this paper is clear. The experiments are also comprehensive and sufficient that can validate the effectiveness of FreeDyG.

**Strengths:**

S1. The overall presentation of this paper is clear, which is easy to grasp the key ideas.

S2. Using the frequency-based Fourier transform to capture the evolving patterns of dynamic graphs is novel and interesting.

S3. The authors conducted comprehensive and sufficient experiments covering both transdutive and inductive settings of temporal link prediction.

**Weaknesses:**

**W1. From my perspective, some of the motivations regarding model designs need further verification or validation.**

In Section 1, the authors argued that RW, TPP, and ODE are computationally expensive. However, the proposed method includes a sampling procedure that samples $L$ first-hop historical neighbors for both source and target nodes. It seems that such a sampling procedure has a complexity similar to those of conventional methods (e.g., RW-based and PPT-based), according to my background knowledge. To verify this motivation, is is recommended to add the comparison of time complexity for sampling/feature extraction in both training and testing phases.

It is also suggested to add pseudocode of each procedure (e.g., first-hop historical neighbors sampling, FFT, extraction of node interaction frequency, etc.) even in the appendix since the details of some modules in current version of manuscript are still unclear.

Moreover, the authors claimed that self-attention acts as persistent low-pass filter and the utilization of DFT can tackle its limitation. How this superiority of the proposed method is validated in the experiments?

***

**W2. As stated in Section 2, the authors only considered CTDG with edge addition events. It seems that the proposed method cannot handle the deletion of edges.**

***

**W3. It semes that there are some inconsistent and unclear statements.**

In Eq. (1) $n$ starts from 0 but in the 2nd paragraph of Section 2, the authors defined that ${x_n}_{n=1}^N$, where $n$ starts from 1. It is also similar for $ {X_k} _{k=1}^N$.

In the 2nd paragraph of Section 3.1, the definitions of $\alpha$ and $\beta$ are not given.

In Eq. (11), what is the dimensionality setting of $W^{agg}$? It is still unclear how to derive a vector $h_*^t$ based on a matrix $Z_*^l$. Moreover, there is no $t$ in the right side of Eq. (11) but how can we know $t$ in the left side?

***

**W4. There are also some minor errors.**

e.g., 'In addition, We specifically encode' > 'In addition, we specifically encode'

**Questions:**

According to my background knowledge, a significant property of CTDG is that the difference between two successive time steps can be irregular. However, as shown in Table 4, each dataset has an item 'Duration'. What does this item mean? Does is mean that the time steps of all the datasets are still regularly spaced?

  In some previous studies, the inductive settings include the prediction between (i) one previously observed node and one newly added node as well as (ii) between two new nodes. It is unclear that the inductive setting in this study refers to which case?

  According to my understanding, the inductive inference of the proposed method and other baselines relies on the availability of graph attributes (i.e., node and edge attributes in this study). Consider an extreme case, when attributes are unavailable, can the proposed method still support the inductive temporal link prediction?

  In addition to the commonly used settings of temporal link prediction in this study (i.e., the prediction of unweighted feature links), there are some other studies considered the advanced temporal link prediction tasks for weighted dynamic graphs [1-4], which should not only determine the existence of a future link but also the corresponding edge weight. Can the proposed method be extended to handle such an advanced settings?

  [1] GCN-GAN: A Non-linear Temporal Link Prediction Model for Weighted Dynamic Networks. IEEE InfoCOM, 2019.

  [2] An Advanced Deep Generative Framework for Temporal Link Prediction in Dynamic Networks. IEEE IEEE Transactions on Cybernetics, 2020.

  [3] High-Quality Temporal Link Prediction for Weighted Dynamic Graphs via Inductive Embedding Aggregation. IEEE TKDE, 2023.

  [4] Temporal link prediction: A unified framework, taxonomy, and review. ACM Computing Surveys, 2023.

---

> ### Author Response · Authors · 2023-11-22
>
> Thank you for your constructive comments and suggestions, and they are exceedingly helpful for us to improve our paper. We have carefully incorporated them in the revised paper.
> In the following, your comments are first stated and then followed by our point-by-point responses.
>
> **W1. From my perspective, some of the motivations regarding model designs need further verification or validation.**
>
> Response: Thank you for your comments. We give our point-by-point responses as follows:
>
> (1) The FreeDyG model employs a sampling strategy that differs from other methods. Specifically, it involves directly sampling a fixed number of first-order neighbors from the historical interactions of the target node, prioritizing their temporal proximity. This is achieved by selecting the $L$ most recent neighbors from the historical interactions, a process with a time complexity of $O(1)$.
> As a result, the sampling method used in FreeDyG is more efficient than RW-based and TPP-based.
>
> (2) We have added the pseudocode of our algorithm in Appendix D of the revised paper. Please refer to Algorithm 1 and Algorithm 2.
>
>
> (3) Further experiments are carried out to demonstrate the efficiency of our filter layer.
> Specifically, we replace the frequency-enhanced MLP-Mixer layer with Transformer and RNN models to compare their performances, as detailed in the following table.
> Following reviewer Gxoa's recommendation, we employ the micro-F1 score as the evaluation metric.
> The results reveal that our FreeDyG model, equipped with the filter layer, delivers the highest performance across all datasets.
> |              | Wiki                | Reddit              | MOOC                | LastFM              | Enron               | Social Evo          | UCI                  |
> |--------------|---------------------|---------------------|---------------------|---------------------|---------------------|---------------------|----------------------|
> | **FreeDyG**  | **95.92 ± 0.09**    | **96.03 ± 0.07**    | **76.74 ± 0.97**    | **88.89 ± 0.38**    | **88.91 ± 0.20**    | **93.87 ± 0.07**    | **90.25± 0.08**      |
> | Transformer  | _94.83 ± 0.13_      | _95.89 ± 0.14_      | _74.89 ± 1.20_      | _87.57 ± 0.38_      | 87.01 ± 0.39        | _92.86 ± 0.24_      | _88.32± 0.64_        |
> | RNN          | 93.94 ± 0.31        | 95.03 ± 0.20        | 74.31 ± 1.18        | 87.19 ± 0.75        | _87.74 ± 0.61_      | 92.23 ± 0.29        | 87.85± 0.58          |
>
>
> **W2. As stated in Section 2, the authors only considered CTDG with edge addition events. It seems that the proposed method cannot handle the deletion of edges.**
>
> Response: Thank you for your comment.
> Our method FreeDyG currently can not handle the deletion of edges. As described in the Conclusion, we plan to expand our research to include edge deletion in future works.
>
> **W3. It seems that there are some inconsistent and unclear statements**
>
> Response: Thank you for your feedback.
> In the revised paper, we have addressed and corrected all inconsistencies and ambiguities. The adjustments made are as follows:
>
> (1) In Eq. (1), we have modified the starting point of $n$ to begin from $1$.
>
> (2) The hyperparameters $\alpha$ and $\beta$ are set to ensure that $t_{max} \times \alpha^{-(i-1) / \beta}$ approaches 0. The time encoding method used in our study is sourced from GraphMixer, as detailed in their publication available at [https://arxiv.org/pdf/2302.11636.pdf].
> For more comprehensive information on this, please refer to the GraphMixer documentation.
> In our experiments, we have chosen the values of both $\alpha$ and $\beta$ to be 10.
>
> (3) The parameter $\mathbf{W^{agg}}\in \mathbb{R}^{1 \times L}$ is a trainable parameter that is designed to adaptively determine the significance of various interactions.
> With $\mathbf{W^{agg}}$, we transform matrix $Z_*^l$ into vector $h_*^t$ with Equ. (11).
> Additionally, the inconsistency in the superscript $t$ arises from its omission in Section 3.1 for simplicity's sake.
> Those $Z_*^l$ is short for $Z_*^{l,t}$.
> We apologize for any confusion this may have caused.

---

> > ### Author Response · Authors · 2023-11-22
> >
> > **Q1. According to my background knowledge, a significant property of CTDG is that the difference between two successive time steps can be irregular. However, as shown in Table 4, each data set has an item 'Duration'. What does this item mean? Does it mean that the time steps of all the data sets are still regularly spaced?**
> >
> > Response: Thank you for your question.
> > In Table 4, 'Duration' refers to the span of time over which the dataset's data was gathered or took place. 'Time Granularity' denotes that each interaction in the dataset is timestamped using Unix timestamps. For example, in the WIKI dataset, the interval from the initial to the final interaction spans approximately one month.
> > And the timestamp for the first interaction is marked as 0 seconds. The subsequent interactions are recorded at intervals of 36 seconds, 77 seconds, 131 seconds, 150 seconds, 153 seconds, and so forth.
> >
> > **Q2: In some previous studies, the inductive settings include the prediction between (i) one previously observed node and one newly added node as well as (ii) between two new nodes. It is unclear that the inductive setting in this study refers to which case?**
> >
> > Response: Thank you for your comments.
> > In our experiments, the inductive setting is to make predictions between two new nodes, neither of which were observed during the training phase.
> > We have further clarified the setting in Section 4.3 of the revised paper.
> >
> > **Q3: According to my understanding, the inductive inference of the proposed method and other baselines relies on the availability of graph attributes (i.e., node and edge attributes in this study). Consider an extreme case, when attributes are unavailable, can the proposed method still support the inductive temporal link prediction?**
> >
> > Response: Thank you for your question.
> > Table 4 shows that all datasets used in the experiments are devoid of node features, for which we have employed zero vectors as node encodings.
> > Additionally, edge features are missing in three datasets (lastfm, enron, uci), aligning with the scenario mentioned. Nonetheless, our approach effectively leverages alternative data, such as temporal details, the proportion of common neighbor nodes, and interaction frequency between node pairs. This additional information aids significantly in facilitating inductive temporal link prediction, despite the absence of node/edge features.
> >
> >
> > **Q4:In addition to the commonly used settings of temporal link prediction in this study (i.e., the prediction of unweighted feature links), there are some other studies considered the advanced temporal link prediction tasks for weighted dynamic graphs [1-4], which should not only determine the existence of a future link but also the corresponding edge weight. Can the proposed method be extended to handle such advanced settings?**
> >
> > Response:
> > Thank you for your insightful questions.
> > Link prediction for weighted dynamic graphs is also an important problem.
> > Our FreeDyG model has the potential to be adapted to address this problem by modifying the encoder layer and the loss function. Specifically, incorporating edge weights into the encoder layer and transitioning from cross-entropy loss to mean squared error (MSE) loss could be effective strategies. Unfortunately, due to time constraints, these experiments are still in progress. In upcoming work, we aim to develop a more efficient encoder designed to effectively capture weighted information within dynamic graphs.

---

### Official Review · Reviewer_VGzm · 2023-10-31

**Soundness:** 3 good
**Presentation:** 3 good
**Contribution:** 3 good
**Rating:** 8
**Confidence:** 3

**Summary:**

This paper introduces a new GNN for CTDG. The concept of Node Interaction Frequency (NIF) Encoding appears to be a simplified version of SEAL, a link prediction technique for static graphs, and it further introduces a frequency-enhanced MLP-Mixer layer that has functions of Fourier transform and inverse transform with weight learning. Evaluation is conducted in various experimental settings, including transductive/inductive and three negative sampling strategies.

**Strengths:**

S1. The overall architecture is well-designed. Of particular interest, the Node Interaction Frequency (NIF) Encoding and frequency-enhanced MLP-Mixer layer are novel and highly effective.

S2. The proposal achieves performance higher than the state-of-the-art. Particularly, achieving high efficiency and high quality is impressive. The ablation study verifies that each technical component is effective for high accuracy.

S3. The experimental settings are detailed, encompassing evaluation experiments across 9 methods, 7 real-world datasets, and various settings including transductive/inductive and three negative sampling strategies.

**Weaknesses:**

W1. Since the proposal's effectiveness varies across datasets, it is essential to discuss the impact of Node Interaction Frequency (NIF) Encoding and the frequency-enhanced MLP-Mixer layer by investigating the characteristics of each dataset. For instance, if a certain dataset is known to exhibit periodicity, it would be reasonable to understand the benefits of the frequency-enhanced MLP-Mixer layer. Similarly, an analysis should be conducted to determine which data characteristics justified the effectiveness of NIF Encoding.

W2. Some design decisions are not clear. For example, while it is crucial that F^t_* represents common neighbors and their past interactions, further explanation is needed regarding the idea behind Equation 3.

W3. The equation transformation involving w_k^{(t)} in Equation (9) is not clear. Additional clarification is necessary to understand this transformation.

**Questions:**

Q1. It would be valuable to discuss how the proposal outperforms other approaches, such as using RNN or transformers, which are known to capture some temporal patterns. This comparison can provide insights into the superior performance of the proposal.

Q2. Could you please clarify whether the optimization is conducted in an end-to-end fashion?

Q3. What is the mean of the circle sizes in Figure 2?

---

> ### Author Response · Authors · 2023-11-22
>
> Thank you for your constructive comments and suggestions, and they are exceedingly helpful for us to improve our paper. We have carefully incorporated them in the revised paper.
> In the following, your comments are first stated and then followed by our point-by-point responses.
>
> **W1: Since the proposal's effectiveness varies across datasets, it is essential to discuss the impact of Node Interaction Frequency (NIF) Encoding and the frequency-enhanced MLP-Mixer layer by investigating the characteristics of each dataset. For instance, if a certain dataset is known to exhibit periodicity, it would be reasonable to understand the benefits of the frequency-enhanced MLP-Mixer layer. Similarly, an analysis should be conducted to determine which data characteristics justified the effectiveness of NIF Encoding.**
>
> Response: Thank you for your question.
> The ablation study experiments depicted in Figure 3 were conducted under the random negative sampling setting, where the target node of a negative edge is sampled from the entire graph. In this setting, the negative samples are significantly easier to distinguish, and the model tends to predict future interactions between node pairs that have interacted previously. Consequently, the NIF encoding, which captures neighbor interactions, assumes a more critical role.
>
> Furthermore, the following table presents additional experimental results from the ablation study conducted using the historical negative sampling setting. In this setting, the target node of a negative edge is sampled from the source node's historical neighbors. It is evident that the FE component holds greater significance in this setting. The reason behind this is that in this setting, the NIF encoding may introduce more false positives on negative samples, necessitating the importance of FE in capturing temporal pattern or shifting signals. For more detailed information, please refer to Figure 5 in the revised version of the paper.
> |              | Wiki                  | Reddit               | MOOC                 | LastFM               | Enron                | Social Evo           | UCI                   |
> |--------------|-----------------------|----------------------|----------------------|----------------------|----------------------|----------------------|-----------------------|
> | **FreeDyG**      | **91.59 ± 0.57**      | **85.67 ± 1.01**     | **86.71 ± 0.81**     | **79.71 ± 0.51**     | **78.87 ± 0.82**     | **97.79 ± 0.23**     | **86.10± 1.19**       |
> | w/o NIF      | 90.01 ± 0.21          | 82.51 ± 1.22         | 84.64 ± 1.13         | 76.61 ± 0.50         | 77.28 ± 1.29         | 96.01 ± 0.30         | 85.11± 1.07           |
> | w/o FE       | 87.31 ± 0.31          | 81.79 ± 0.75         | 85.01 ± 1.18         | 76.54 ± 0.62         | 75.74 ± 0.61         | 96.43 ± 0.20         | 83.09± 1.54           |
>
> **W2.Some design decisions are not clear. For example, while it is crucial that $F^t_*$ represents common neighbors and their past interactions, further explanation is needed regarding the idea behind Equation 3.**
>
> Response: Thank you for your comments.
> In our study, we utilize $F^t_u$ to denote the historical common neighbors between node $u$ and $v$. Diverging from previous approaches that only calculate the count of common neighbors, our model defines $F^t_u$ as a 2-dimensional vector aimed at capturing the temporal patterns concealed within these common neighbors.
> The motivation of Equation 3 is straightforward. We use a sum pooling operation to consolidate the count information (which is mapped to a vector with dimension $d_F$ by function $f$) pertaining to common neighbors of nodes $u$ and $v$ and interaction frequency between nodes $u$ and $v$.
>
> **W3. The equation transformation involving $w_k^{(t)} $in Equation (9) is not clear. Additional clarification is necessary to understand this transformation.**
>
> Response: Thank you for your feedback.
> In Equation (9), we have $w_k^{(t)} = \sum_{m=1}^{N} h_m^{(t)} e^{-\frac{2 \pi i}{N} k m}$.
> The description has been incorporated into the revised paper.

---

> > ### Author Response · Authors · 2023-11-22
> >
> > **Q1. It would be valuable to discuss how the proposal outperforms other approaches, such as using RNN or transformers, which are known to capture some temporal patterns. This comparison can provide insights into the superior performance of the proposal.**
> >
> > Response:  Thank you for your suggestion. As we mentioned in Section 1, self-attention mechanism acts as a persistent low-pass filter, treating high-frequency information as noise and continuously erasing it. While effective in certain datasets, it may overlook crucial temporal dynamics in datasets where high-frequency components carry significant information. Though RNN is capable of handling long-term temporal dependencies, it also falls short in effectively processing the full spectrum of frequency information. However, the filter layer in FreeDyG particularly addresses the need for a more nuanced treatment of temporal information, allowing for the retention and effective utilization of high-frequency components that are crucial in dynamic graphs.
> > Further experiments are carried out to demonstrate the efficiency of our filter layer.
> > Specifically, we replace the Frequency-enhanced MLP-Mixer layer with Transformer and RNN models to compare their performances, as detailed in the following table.
> > Following reviewer Gxoa's recommendation, we employ the micro-F1 score as the evaluation metric.
> > The results reveal that our FreeDyG model, equipped with the filter layer, delivers the highest performance across all datasets.
> > |              | Wiki                | Reddit              | MOOC                | LastFM              | Enron               | Social Evo          | UCI                  |
> > |--------------|---------------------|---------------------|---------------------|---------------------|---------------------|---------------------|----------------------|
> > | **FreeDyG**  | **95.92 ± 0.09**    | **96.03 ± 0.07**    | **76.74 ± 0.97**    | **88.89 ± 0.38**    | **88.91 ± 0.20**    | **93.87 ± 0.07**    | **90.25± 0.08**      |
> > | Transformer  | _94.83 ± 0.13_      | _95.89 ± 0.14_      | _74.89 ± 1.20_      | _87.57 ± 0.38_      | 87.01 ± 0.39        | _92.86 ± 0.24_      | _88.32± 0.64_        |
> > | RNN          | 93.94 ± 0.31        | 95.03 ± 0.20        | 74.31 ± 1.18        | 87.19 ± 0.75        | _87.74 ± 0.61_      | 92.23 ± 0.29        | 87.85± 0.58          |
> >
> > **Q2. Could you please clarify whether the optimization is conducted in an end-to-end fashion?**
> >
> > Response:  The optimization is conducted in an end-to-end manner and we add the pseudocode of our algorithm in Appendix D of revised paper. Please refer to Algorithm 1. We will release the code in the future.
> >
> > **Q3. What is the mean of the circle sizes in Figure 2?**
> >
> > Response: Figure 2 illustrates the parameter size of each model using the size of circles. More specifically, the diameter of each circle is in linear proportion to the size of the parameters.

---

> > ### Comment · Reviewer_VGzm · 2023-11-23
> >
> > The authors' response clarifies all of my concerns.

---

> > > ### Author Response · Authors · 2023-11-23
> > >
> > > We are delighted to see that the major concerns raised by the reviewer have been successfully addressed. We would like to express our sincere gratitude to the reviewer for your meticulous examination of our paper and for providing invaluable feedback.

---

### Official Review · Reviewer_Gxoa · 2023-11-01

**Soundness:** 3 good
**Presentation:** 4 excellent
**Contribution:** 2 fair
**Rating:** 6
**Confidence:** 5

**Summary:**

This paper proposes a novel method called FreeDyG for link prediction in dynamic graphs. The method devised a novel frequency-enhanced MLP-Mixer layer to learn the periodic temporal patterns and the ”shift” phenomenon present in the frequency domain. The effectiveness of the FreeDyG model was validated on several real-world datasets, showing performance improvement in AUC-ROC against baselines.

**Strengths:**

1. The proposed frequency-enhanced MLP-Mixer is novel and effective.
2. The experiments of link prediction are comprehensive. It is conducted on seven datasets and compares the performance against 9 baselines in two dynamic settings, which is solid and comprehensive to validate the effectiveness of FreeDyG in link prediction.
3. The paper is well written, especially the problem formulation and methodologies.

**Weaknesses:**

1. The motivation of delving into the frequency domain needs to be further clarified. I am wondering about the intuitions behind capturing the ”shift” phenomenon hidden in the frequency domain.
2. The authors claim that FreeDyG is the first work that considers the frequency information for dynamic graph embedding, which is overclaimed.
3. The authors argue that random walk based approaches are computationally expensive. However, conducting Fourier transform are also very computationally expensive. In addition, I think FreeDyG also relied on some random walk based approach to obtain the continuous-time dynamic graph from the raw graph data.
4. The proposed FreeDyG seems computationally expensive. However, there is no time complexity analysis. The authors are suggested to present the time complexity empirically or theoretically.
5. In a dynamic graph, some nodes will have more edges, but others will have fewer. Using AUC-ROC as the evaluation metrics cannot tell how good the performance of link prediction for minority nodes. I suggest reporting the Micro- and Macro-F1 scores in the link prediction tasks.

**Questions:**

1. Why does this work only focus on the link prediction? How is applying this work applicable to other graph mining tasks like node classifications?
2. For the LastFM, what is the summit of the performance when sampling more neighbor nodes? Could you please clarify the experiment details about training every baseline using the same amount of information as FreeDyG in the experiments?

---

> ### Author Response · Authors · 2023-11-22
>
> Thank you for your constructive comments and suggestions, and they are exceedingly helpful for us to improve our paper. We have carefully incorporated them in the revised paper.
> In the following, your comments are first stated and then followed by our point-by-point responses.
>
> **W1:The motivation of delving into the frequency domain needs to be further clarified. I am wondering about the intuitions behind capturing the 'shift phenomenon hidden in the frequency domain.**
>
> Response: Thank you for your suggestion.
> In the field of time series analysis, examining the frequency domain has become a popular method for effectively capturing temporal signals over time. Similarly, continuous-time graph data also display temporal signals, such as the occurrence of 'shift' phenomena. These phenomena involve significant changes in the interaction patterns of nodes at specific moments. These changes in the connection patterns often result in outlier values or sudden variations in the representation vector $Z^t_*$. The basic idea of our model is straightforward: we aim for our adaptive frequency filter to capture the frequency patterns associated with temporal outliers or sudden changes.
>
>
> **W2:The authors claim that FreeDyG is the first work that considers the frequency information for dynamic graph embedding, which is overclaimed.**
>
> Response: Thank you for pointing out this problem. We have reorganized our statement in the revised paper as follows: *Instead of the temporal domain, FreeDyG tries to address this problem by delving into the frequency domain*.
>
>
> **W3:The authors argue that random walk based approaches are computationally expensive. However, conducting Fourier transform is also very computationally expensive. In addition, I think FreeDyG also relied on some random walk based approach to obtain the continuous-time dynamic graph from the raw graph data.**
>
> Response: Thank you for your comments.
>
> **Time complexity of Fourier transform:** The time complexity of the Fast Fourier Transform (FFT) is given by $O(L \log L)$, where $L$ represents the number of sampled neighbors from historical interactions. When it comes to random walk-based methods such as CAWN, the primary computational cost lies not in the sampling process, but rather in how they utilize these walks.
> Specifically, after getting multiple walks, CAWN uses RNN to encode each walk as a representation and uses self-attention to aggregate the representations of multi-walks into a single vector for downstream tasks. In Figure 2, we present a comparison between our approach, FreeDyG, and the random walk-based method CAWN. It is worth noting that the training time required for CAWN on Wikipedia is double that of FreeDyG. More specifically, FreeDyG and CAWN take $67$ and $150$ seconds per training epoch, respectively. Furthermore, when applied to larger datasets such as Reddit, CAWN incurs a significantly higher time cost in comparison to FreeDyG.
>
> **Sampling strategy of FreeDyG:** The FreeDyG model employs a sampling strategy that differs from random walk-based approaches. Specifically, it involves directly sampling a fixed number of first-order neighbors from the historical interactions of the target node, prioritizing their temporal proximity. This is achieved by selecting the $L$ most recent first-hop neighbors from the historical interactions, a process with a time complexity of $O(1)$.
>
> **Raw graph data:** The continuous-time dynamic graph is defined as a chronological sequence of interactions between specific node pairs as explained in Section 2.
> For our experiments, the raw graph data is also organized in the form of an edge list, consisting of source nodes, target nodes, and timestamps. As a result, there is no need for additional operations to derive the continuous-time dynamic graph from the raw graph data.

---

> ### Author Response · Authors · 2023-11-22
>
> **W4: The proposed FreeDyG seems computationally expensive. However, there is no time complexity analysis. The authors are suggested to present the time complexity empirically or theoretically.**
>
> Response: Let $L$ represent the number of sampled neighbors and $d$ denote the dimension of the embdedding size. Our FreeDyG model comprises several components, each with its respective time complexity. The sampling process, which acquires the $L$ most recent neighbors, exhibits a time complexity of $O(1)$, while the encoding module operates at a time complexity of $O(Ld)$. The time complexity of the filter layer is determined by the FFT and IFFT operations, which amount to $O(L \log L)$. Additionally, the MLP-Mixer layer has a time complexity of $O(Ld)$. Consequently, the overall time complexity of FreeDyG is at most $O(L \log Ld)$. We note that the time complexity of our FreeDyG is lower than the self-attention with $O(L^2 d)$.
> Figure 2 in our experimental analysis displays the performance of all competing methods, along with their corresponding training time and parameter size. Notably, our FreeDyG outperforms all other approaches, achieving the highest performance while maintaining a moderate training cost.
>
> **W5:In a dynamic graph, some nodes will have more edges, but others will have fewer. Using AUC-ROC as the evaluation metrics cannot tell how good the performance of link prediction for minority nodes. I suggest reporting the Micro- and Macro-F1 scores in the link prediction tasks.**
>
> Response: Thank you for your suggestion.
> The results for Micro-f1 and Macro-f1 scores across all competitors are presented in Tables 7-10 in revised paper. These outcomes align with the findings from AP and AUC-ROC measurements. Notably, our model FreeDyG demonstrates superior performance compared to other baseline models in a majority of the test cases. The average ranking of FreeDyG approximates to 1 in most settings, significantly outstripping its closest competitor, DyGFormer.
>
> **Q1.Why does this work only focus on the link prediction? How is applying this work applicable to other graph mining tasks like node classifications?**
>
> Response: Thank you for your comments.
> In this paper, our primary focus is on the task of link prediction using the proposed method. Nonetheless, by altering the task layer and loss function settings, it can also be adapted for node classification tasks. We have performed experiments for node classification on the Wiki dataset with metric AUC-ROC (experiments on other datasets are still running). The results indicate that our method outperforms most baseline models.
> |             | **JODIE**         | DyRep            | TGAT             | TGN              | CAWN             | TCL              | GraphMixer       | DyGFormer        | FreeDyG          |
> |-------------|-------------------|------------------|------------------|------------------|------------------|------------------|------------------|------------------|------------------|
> | **Wiki**    | **88.78 ± 1.13**  | 86.35 ± 1.27 | 84.43 ± 1.46 | 86.27 ± 2.09  | 85.41 ± 1.51  | 76.91 ± 1.99  | 86.59 ± 0.88 | 87.44 ± 1.08     | _87.87 ± 1.13_   |
>
> **Q2.For the LastFM, what is the summit of the performance when sampling more neighbor nodes? Could you please clarify the experiment details about training every baseline using the same amount of information as FreeDyG in the experiments?**
>
> Response:  Thank you for your comments. We have conducted experiments with Micro-F1 metric on LastFM with $L=10,20,32,64,100,128, 160, 200$ in the following table.
> We can see that LastFM exhibits optimal performance when $L=100$.
> This requirement for a larger number of historical neighbors in LastFM is attributed to its higher average node degree, quantified as $|\mathcal{E}|/|\mathcal{V}|=653$.
> which is considerably higher compared to other datasets such as Reddit ($61$), Wiki ($17$), and MOOC ($57$).
> |        | L=10           | L=20           | L=32           | L=64           | L=100          | L=128          | L=160          | L=200          |
> |--------|----------------|----------------|----------------|----------------|----------------|----------------|----------------|----------------|
> | LastFM | 81.28 ± 0.71   | 84.63 ± 0.43   | 88.06 ± 0.49   | 87.76 ± 0.44   | **88.89 ± 0.38** | 88.84 ± 0.39   | 88.27 ± 0.51   | 88.20 ± 0.42   |

---

> > ### Comment · Reviewer_Gxoa · 2023-11-23
> >
> > Thank you for providing a more comprehensive evaluation which is very helpful to understand the usefulness of the proposed FreeDyG. I am raising my score to 6.

---

> > > ### Author Response · Authors · 2023-11-23
> > >
> > > We are delighted to see that the major concerns raised by the reviewer have been successfully addressed. We would like to express our sincere gratitude to the reviewer for your meticulous examination of our paper and for providing invaluable feedback.

---

### Official Review · Reviewer_Wgp4 · 2023-11-03

**Soundness:** 3 good
**Presentation:** 3 good
**Contribution:** 3 good
**Rating:** 8
**Confidence:** 3

**Summary:**

The authors propose the FreeDyG graph neural network (GNN) model for continuous-time dynamic graphs. It incorporates frequency-based representations of the nodes to attempt to capture periodic patterns in the dynamic graph. It also contains a novel node interaction frequency encoding approach. The authors demonstrate impressive link prediction accuracy in both transductive and inductive settings compared to other GNNs for dynamic graphs on a variety of real data sets. They also perform favorably in terms of training time and size of the trained model when compared to other methods.

*After author rebuttal:* The authors have partially addressed my concerns in their rebuttal and revision, particularly regarding the usefulness of the frequency encoding. After reading through the other reviews, I am still in support of this paper.

**Strengths:**

- Proposed FreeDyG model contains several novel elements, including the design of the node interaction frequency (NIF) encoding and incorporating frequency-based representations.
- Comparison of accuracy, training time, and model size shown in Figure 2 is a nice inclusion. This shows that improvements in accuracy are not at the cost of significantly increased training time or a very large model.
- Strong improvements in accuracy compared to other approaches. These improvements hold over different negative sampling strategies and evaluation metrics.
- Mostly well written and organized paper. In my opinion, the authors made good choices on which results and details should be presented in the main paper rather than the appendices.

**Weaknesses:**

- The positioning of the paper is a bit deceiving. From reading the paper, it would appear as though the main contribution is incorporating the frequency information. However, from the results of the ablation study in Figure 3, we see that the NIF encoding plays a much bigger role in improving accuracy than the frequency-based representations.
- The authors present no evidence that the frequency-based representations are actually able to capture periodic patterns, which they used as their motivation for using the FFT.

Minor concerns:
- Table 3 is probably not the best way to present the hyperparameter study. Typically, one would be looking for trends as you vary the number of historical neighbors $L$. Such trends are difficult to pick out from the table. I would suggest instead using plots with AP or AUC-ROC on one axis and $L$ on the other.
- Page 7, second last paragraph: "neigh encoding"

**Questions:**

1. Is there a way you could inspect your trained model to identify whether any type of periodic patterns are being captured by your frequency-enhanced MLP-Mixer layer?
2. Why is the NIF encoding more important than the frequency-based representations for improving link prediction accuracy?

---

> ### Author Response · Authors · 2023-11-22
>
> Thank you for your constructive comments and suggestions, and they are exceedingly helpful for us to improve our paper. We have carefully incorporated them in the revised paper.
> In the following, your comments are first stated and then followed by our point-by-point responses.
>
> **Concerns 1: Table 3 is probably not the best way to present the hyperparameter study. Typically, one would be looking for trends as you vary the number of historical neighbors. Such trends are difficult to pick out from the table. I would suggest instead using plots with AP or AUC-ROC on one axis and on the other.\***
>
> Response: Thank you for your suggestion. In Appendix D of the revised paper, we use Figure 4 to show the trends of AP along different numbers of sampled historical first-hop neighbors.
>
> **Concerns 2: Page 7, second last paragraph: "neigh encoding"**
>
> Response: Thank you for pointing out the typo error, we have fixed it in the revised paper.
>
> **Q1: Is there a way you could inspect your trained model to identify whether any type of periodic patterns are being captured by your frequency-enhanced MLP-Mixer layer?**
>
> Response: Thank you for your comments. The frequency-enhanced MLP-Mixer layer is used to capture the periodic and shift patterns among the representation space of the historical first-hop neighbors.
> Different from the time series analysis problem, the periodic patterns in the hidden space of a graph may be caused by multiple interaction patterns between nodes such as repeated interactions with the same node, similar frequency of interactions, or even high-order statistic repetition.
>
> **Q2: Why is the NIF encoding more important than the frequency-based representations for improving link prediction accuracy?**
>
> Response: Thank you for your question.
> The ablation study experiments depicted in Figure 3 were conducted under the random negative sampling setting, where the target node of a negative edge is sampled from the entire graph. In this setting, the negative samples are significantly easier to distinguish, and the model tends to predict future interactions between node pairs that have interacted previously. Consequently, the NIF encoding, which captures neighbor interactions, assumes a more critical role.
>
> Furthermore, the following table presents additional experimental results from the ablation study conducted using the historical negative sampling setting. In this setting, the target node of a negative edge is sampled from the source node's historical neighbors. It is evident that the FE component holds greater significance in this setting. The reason behind this is that in this setting, the NIF encoding may introduce more false positives on negative samples, necessitating the importance of FE in capturing temporal or shifting signals. For more detailed information, please refer to Figure 5 in the revised version of the paper.
> |              | Wiki                  | Reddit               | MOOC                 | LastFM               | Enron                | Social Evo           | UCI                   |
> |--------------|-----------------------|----------------------|----------------------|----------------------|----------------------|----------------------|-----------------------|
> | **FreeDyG**      | **91.59 ± 0.57**      | **85.67 ± 1.01**     | **86.71 ± 0.81**     | **79.71 ± 0.51**     | **78.87 ± 0.82**     | **97.79 ± 0.23**     | **86.10± 1.19**       |
> | w/o NIF      | 90.01 ± 0.21          | 82.51 ± 1.22         | 84.64 ± 1.13         | 76.61 ± 0.50         | 77.28 ± 1.29         | 96.01 ± 0.30         | 85.11± 1.07           |
> | w/o FE       | 87.31 ± 0.31          | 81.79 ± 0.75         | 85.01 ± 1.18         | 76.54 ± 0.62         | 75.74 ± 0.61         | 96.43 ± 0.20         | 83.09± 1.54           |

---

### Author Response · Authors · 2023-11-22

We thank all the reviewers for their insightful and constructive feedback. We really appreciate all four reviewers thought our method to be "novel" and "have good empirical results".

We have made our point-to-point response to the comments of each reviewer and uploaded our revised version. Finally, we once again thank all reviewers for their insightful comments which are very helpful for improving the quality of our paper.

---

### Comment · Area_Chair_YSgr · 2023-11-22
**Less than one day**

Dear Reviewers,

If you have already responded to authors rebuttal, Thank you!
If not, please take some time, read their responses and acknowledge by replying to the comment. Please also update your score, if applicable.

Thanks everyone for a fruitful, constructive, and respectful review process.

Cheers, Your AC!

---

### Meta-Review · Area_Chair_YSgr · 2023-12-10

**Metareview:**

This paper proposes a graph neural network model for link prediction in continuous time dynamic graphs. It introduces a frequency-enhanced MLP layer to capture periodic temporal patterns and shift in the frequency domain. Experiments on real-world datasets demonstrate  outperforming existing methods for link prediction in a variety of seetings.

Strengths:
-  The proposed model introduces a few novel elements for link prediction and interaction modeling, like node interaction frequency encoding and frequency-enhanced MLP-Mixer layer which seem to be of key importance in the success of the method.
- It achieves higher link prediction performance compared to SOTA methods, while maintaining high efficiency.
- The paper provides comprehensive experiments are conducted, including comparisons across different  methods and datasets, transductive/inductive settings, and negative sampling strategies.

Weaknesses:
- The frequency domain is usually hard to make a sene. The motivation and intuitions behind capturing the shift should be elaborated more.

Rooms for improvement:
- More detailed analysis and intuition behind the design decisions, especially the frequency domain aspects.

**Justification For Why Not Higher Score:**

Lack of elaboration and justification on the intuitions behind the frequency domain and design decisions.

**Justification For Why Not Lower Score:**

The paper is very well written with solid results and novel techniques.

---

### Decision · Program_Chairs · 2024-01-16

Accept (poster)